# Impact of airborne cloud radar reflectivity data assimilation on kilometre-scale NWP analyses and forecasts of heavy precipitation events

Mary Borderies[1], Olivier Caumont[1], Julien Delanoë[2], Véronique Ducrocq[1], Nadia Fourrié[1], and Pascal Marquet[1]

[1]CNRM, Université de Toulouse, Météo-France, CNRS, Toulouse, France
[2]LATMOS, IPSL, Université Versailles St-Quentin, CNRS, UPMC, Guyancourt, France

**Correspondence:** mary.borderies@meteo.fr

**Abstract.**

This article investigates the potential of W-band radar reflectivity to improve the quality of analyses and forecasts of heavy precipitation events in the Mediterranean area. The 1D+3DVar assimilation method, operationally employed to assimilate ground-based precipitation radar data in the Météo-France kilometre-scale NWP model AROME, has been adapted to assimi-
late the W-band reflectivity measured by the airborne cloud radar RASTA during a two-month period over the Mediterranean area. After applying a bias correction, vertical profiles of relative humidity are first derived via a 1D Bayesian retrieval, and then used as relative humidity pseudo-observations in the 3DVar assimilation system of AROME. The efficiency of the 1D Bayesian method in retrieving humidity fields is assessed using independent in-flight humidity measurements. To complement this study, the benefit brought by consistent thermodynamic and dynamic cloud conditions has been investigated by assim-
ilating separately and jointly in the 3h 3DVar assimilation system of AROME the W-band reflectivity and horizontal wind measurements collected by RASTA.

The data assimilation experiments are conducted for a single heavy precipitation event, and then for 32 cases. Results indicate that the W-band reflectivity has a larger impact on the humidity, temperature and pressure fields in the analyses, compared to the assimilation of RASTA wind data alone. Besides, the analyses get closer to independent humidity observations if the W-
band reflectivity is assimilated alone or jointly with RASTA wind data. Nonetheless, the impact of the W-band reflectivity decreases more rapidly as the forecast range increases, compared to the assimilation of RASTA wind data alone. Generally, the assimilation of the W-band reflectivity jointly with wind data results in the best improvement of the rainfall precipitation forecasts. Consequently, results of this study indicate that consistent thermodynamic and dynamic cloud conditions in the analysis leads to an improvement of both model initial conditions and forecasts. Even though to a less extent, the assimilation of the W-band reflectivity alone also results in a slight improvement of the rainfall precipitation forecasts.

# 1 Introduction

Kilometre-scale NWP models are now able to explicitly resolve the convection and to represent with a reasonable degree of realism clouds and precipitation (Gustafsson et al., 2018). Doppler radar observations are particularly well suited to initialise these NWP models because they provide wind and reflectivity measurements at a comparable spatial and temporal resolution. Consequently, observations from ground-based precipitation radars are operationally assimilated in many km-scale NWP models to initialise precipitation areas (Gustafsson et al., 2018). However, ground-based precipitation radars have a very poor sensitivity to clouds. Hence, the increased number of Doppler cloud-profiling radar (Wolde and Pazmany, 2005; Delanoë et al., 2013; Illingworth et al., 2015; Chahat et al., 2016; Delanoë et al., 2016) is extremely appealing in data assimilation to initialise km-scale NWP models in cloudy areas. Indeed, cloud-profiling radars, either operating in the Ka-band ($\approx$ 35 GHz), or in the W-band ($\approx$ 95 GHz), provide valuable observations about cloud microphysical properties and light to moderate precipitation (Kollias et al., 2007). In addition, compared to lower-frequency radars, they require small antennas to provide high spatial resolution measurements, which makes them more easily deployable aboard moving platforms such as spacecraft or aircraft (Kollias et al., 2007). In addition, recent technological breakthroughs might lead to a deployment of lower-cost ground-based W-band radar networks (Delanoë et al., 2016).

Over the last few years, cloud radar data have been used several times for kilometre-scale NWP model validation (Di Michele et al., 2012; Iguchi et al., 2012; Hashino et al., 2013; Borderies et al., 2018), but only a few studies were devoted to evaluate their potential in data assimilation. In particular, within the JMA's nonhydrostatic model (JMA-NHM) with an ensemble variational method (Aonashi and Eito, 2011), Okamoto et al. (2016) performed a direct assimilation of vertical reflectivity profiles of the Dual frequency Precipitation Radar (DPR) reflectivity observations from the GPM core observatory (Hou et al., 2014) for the case of Typhoon Halong in July 2014. The assimilation of DPR data resulted in an improvement in the rain mixing ratio and updraft. However, because of negative model biases in the ice regions, observations were discarded in and above the melting layer. Therefore, Okamoto et al. (2016) did not take advantage of the cloud-affected observations measured by the Ka band radar, which are very sensitive to clouds. In addition, ensemble variational methods are costly in terms of computing time, which hampers their use in most current operational systems.

Assimilating reflectivity with traditional variational assimilation techniques (3DVar and 4DVar) remains challenging. Indeed, the linearisation of the observation forward operator is not straightforward. In addition, it is necessary to add hydrometeor contents in the control variables, which requires to specify the associated forecast error covariances. Besides, the assimilation of humidity, wind and temperature variables have a larger impact on the forecasts, compared to hydrometeor observations (Fabry and Sun, 2010). Consequently, several studies used indirect assimilation methods to assimilate cloud radar reflectivity measurements (Storto and Tveter, 2009; Janiskovà et al., 2012; Janisková, 2015). The reflectivity profiles are first inverted into pseudo-observations that are closer to the control variables of the NWP model (for instance temperature or humidity) through the use of either a 1D-variational assimilation technique (Janiskovà et al., 2012; Janisková, 2015) or a 1D Bayesian retrieval

(Storto and Tveter, 2009). These pseudo-observations are then assimilated into the variational assimilation system like any other conventional observation. In most of these studies, Cloud Profiling Radar data on board the CloudSat (Stephens, 2005) satellite were assimilated in NWP models running at coarse horizontal resolutions (larger than 10 km). For example, using a technique combining a one dimensional variational (1DVar) followed by a four dimentional variational (4DVar) assimilation method, Janisková (2015) performed several assimilation experiments with the global scale NWP model IFS. To take fully advantage of the W-band reflectivity in cloudy areas, Janisková (2015) applied an appropriate bias correction scheme which depends on the altitude and on the temperature. Results suggest a slight positive impact on the subsequent forecasts when appropriate bias correction, error estimates and quality controls are performed. However, because of the inability of the reflectivity forward operator (Di Michele et al., 2012) to simulate the multiple scattering effects, observations of the most convective situations were rejected from the assimilation process. Storto and Tveter (2009) also employed a two-step method consisting of a one-dimensional Bayesian retrieval of relative humidity pseudo-observations, followed by a 3DVar assimilation method in the ALADIN NWP model (Fischer et al., 2005). Results show that, despite the small number of assimilated observations, the impact of relative humidity pseudo-observations is greater in areas poorly covered by the conventional observation networks, such as over the oceans. However, Storto and Tveter (2009) failed to identify a case study for which the humidity pseudo-observations led to a significant impact on the analysis and on the subsequent forecasts.

So far, the impact of the assimilation of W-band radar reflectivity in a kilometre-scale NWP model, with horizontal resolutions of less than 3 km, has never been investigated. Therefore, the primary objective of this article is to investigate the benefits brought by the assimilation of W-band radar reflectivity measurements to improve the forecasts of the heavy precipitation events that regularly occur in the Mediterranean area. Indeed, the accurate forecasting of the timing, position and intensity of such mesoscale convective systems remains a challenge (Duffourg et al., 2016; Martinet et al., 2017). In addition, Doppler cloud radar data also provide valuable informations on dynamical cloud properties. Borderies et al. (in review) highlighted a positive impact of the assimilation of such data in a km-scale NWP model. The assimilation of W-band reflectivity measurements jointly with wind data measured by Doppler cloud radar is expected to provide more consistent thermodynamic and dynamic cloud conditions in the initial state. Nonetheless, Bachmann et al. (2018) suggested that the joint assimilation leads to skill which are comparable to the experiments in which reflectivity and Doppler velocity observations are assimilated independently. However, their data assimilation experiments were conducted in an idealized setup, and the observations were provided by ground-based precipitation radar data. Therefore, to investigate the benefit brought by consistent thermodynamic and dynamic cloud conditions in the initial state, the W-band reflectivity will be assimilated separately and jointly with horizontal wind data measured by a Doppler W-band radar.

To assess the potential of Doppler W-band radar data to improve short term forecasts of heavy precipitation events, we take advantage of the data collected by the airborne Doppler W-band radar RASTA (Radar Airborne System Tool for Atmosphere Delanoë et al., 2013) in 2012 over the Western Mediterranean area during the HyMeX first Special Observing Period (HyMeX-SOP1 Ducrocq et al., 2014) dedicated to heavy precipitation events. The W-band reflectivity and wind measurements

collected by RASTA are assimilated separately and jointly into the 3DVar assimilation system of a special version, named AROME-WMed (Fourrié et al., 2015), of the Météo-France operational convective scale model AROME (Seity et al., 2011). The impact of the assimilation of RASTA data in synergy with all other conventional assimilated data is first evaluated for one of the most significant rainfall events which occurred during the Intensive Observing Period 7a (Hally et al., 2014, IOP7a) of the HyMeX-SOP1. Next, the experiments are run for 32 case studies of the HyMeX-SOP1 in which RASTA data are available. The 1D+ 3DVar assimilation method of Caumont et al. (2010), used operationally to assimilate ground-based precipitation radars in AROME (Wattrelot et al., 2014), is particularly well suited for vertically pointing radars and is thus employed to assimilate the W-band reflectivity observed by RASTA. Vertical profiles of relative humidity are first derived via a 1D Bayesian retrieval, and then used as pseudo-observations in the 3D-Var assimilation system of AROME. For the first time, a validation of the 1D Bayesian retrieval with humidity in-situ measurement is performed in this study.

This article is organised as follows. In section 2, the data collected by the airborne Doppler W-band radar RASTA are presented, followed by a brief description of the NWP model AROME-WMed with its 3h-3DVar assimilation system. Section 3 provides a full description of the 1D+3DVar assimilation method used to assimilate the W-band radar reflectivity. The different data assimilation experiments are presented in section 4. These different experiments are evaluated in section 5, followed in section 6 by an evaluation over the 32 assimilation cases of the SOP1. Conclusions are drawn in section 7.

## 2   NWP model and radar data

This study takes advantage of the data collected by the Doppler W-band radar RASTA during the HyMeX first special observing period (HyMeX-SOP1), which took place from 5 September to 5 November 2012 over the western Mediterranean (Ducrocq et al., 2014). The main goal of the HyMeX-SOP1 was to document the heavy rainfall events that regularly affect the Mediterranean area.

### 2.1   The Doppler W-band radar RASTA during the HyMeX-SOP1

The airborne cloud radar RASTA is a monostatic Doppler multi-beam antenna system operating in the W-band at 95 GHz (Bouniol et al., 2008, Protat et al., 2009, Delanoë et al., 2013). The aircraft platform used is the French Falcon 20 research aircraft from the SAFIRE unit (Service des Avions Français Instrumentés pour la Recherche en Environnement). RASTA has six Cassegrain antennas to measure the reflectivity and the radial velocity in three non-collinear directions above and below the aircraft. The maximum unambiguous distance is 15 km with an unambiguous velocity of 7.8  m s$^{-1}$ (the Pulse Repetition Frequency equals 10 kHz).

After processing, the Doppler velocities of the three upward-looking and downward-looking antennas are combined to retrieve the three components of the wind field above and below the aircraft (Bousquet et al., 2016). The measurements are collected at a time resolution of 250 ms (i.e. 1.5 s between two measurements of the same antenna) and at a vertical resolution

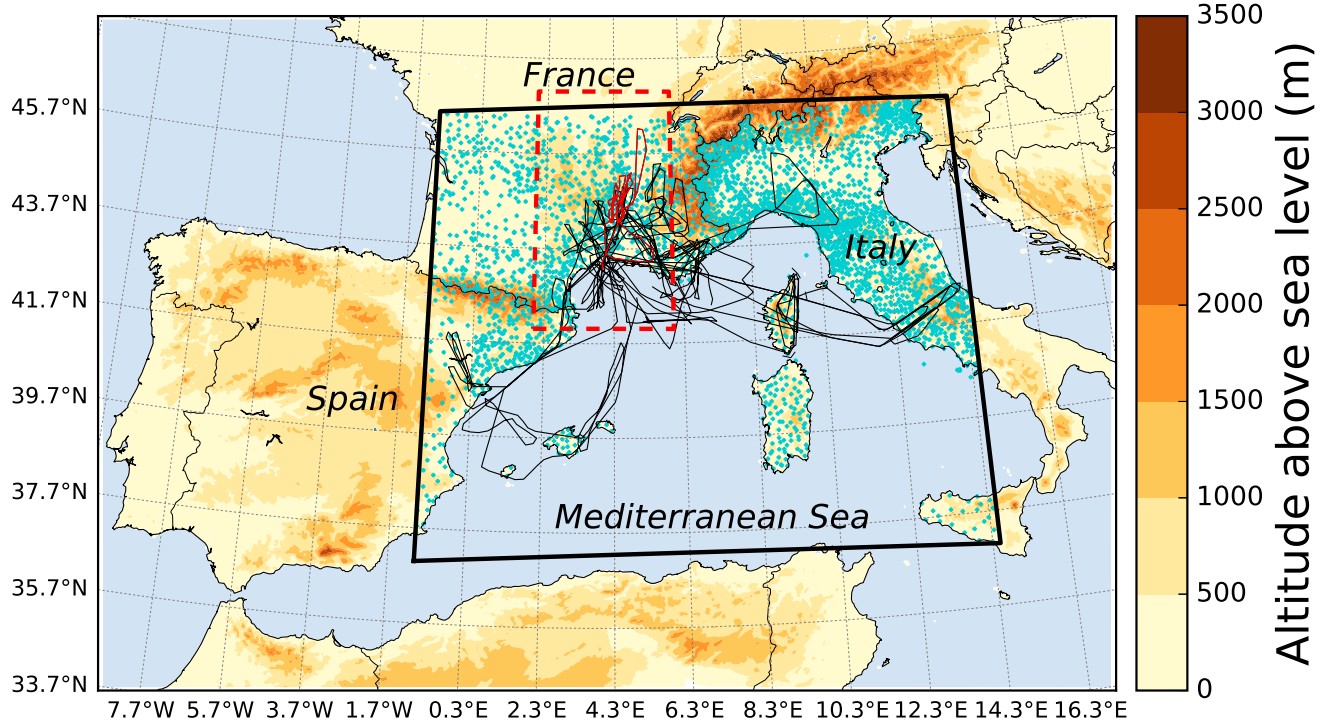

**Figure 1.** The Falcon 20 flight tracks (black lines) during the HyMeX first Special Observing Period over the AROME-WMed domain. The Falcon 20 flight track during IOP7a (Flight 15) is indicated by the red line. IOP7a case study is delimited by the red box. The altitude of ground above sea level (in metres) is represented by the colour shades. Rain-gauges are represented by the blue markers. The black box shows the domain used for the impact study in subsection 6.1.

of 60 m. In addition, this study takes advantage of the reflectivity measurements collected by the nadir- and zenith-pointing antennas. The zenith-pointing antenna is slightly less sensitive than the nadir-pointing antenna (-26 dBZ versus -27 dBZ at 1 km).

5     Therefore, this unique instrument allows the documentation of the microphysical and dynamic properties of clouds in the vertical at a high resolution of 60 m and quasi-continuously in time ($\approx$ 1.5 s) during the flights. In particular, during the HyMeX-SOP1, RASTA collected data during 18 flights in stratiform (72.6%), convective (14.3%) and clear sky (13.1%) *columns* over land, sea and complex terrains (Borderies et al., 2018). RASTA flight tracks during the HyMeX-SOP1 are represented by the black lines in Figure 1. Further details about RASTA configuration during the HyMeX-SOP1 are given by Bousquet et al. 10  (2016).

## 2.2 The AROME-WMed NWP model

This study is performed with a special version of the Météo-France operational kilometre-scale NWP model AROME (Seity et al., 2011), named AROME-WMed (Fourrié et al., 2015). AROME-WMed, which covers the entire northwestern Mediterranean Basin, was specially designed for the HyMeX-SOP1 and ran in real time to plan the airborne operations, especially in the mesoscale convective systems. The AROME-WMed domain is displayed in Figure 1. AROME-WMed runs at a horizontal resolution of 2.5 km with 60 vertical levels ranging from approximately 10 m above ground level to 1 hPa. The deep convection is explicitly resolved and the microphysical processes are governed by the ICE3 one-moment bulk microphysical scheme (Pinty and Jabouille, 1998). Six water species are predicted by AROME-WMed (water vapour, rain, cloud liquid droplets, snow, pristine ice and graupel). The Particle Size Distributions (PSDs) are expressed as generalized gamma distributions multiplied by the total number concentrations. PSDs are reduced to exponential distributions for snow, graupel and rain.

The analyses of the global operational NWP model ARPEGE are used to provide boundary conditions. AROME-WMed has a 3-h 3D Variational (3DVar) data assimilation system (Brousseau et al., 2014) based on an incremental formulation (Fischer et al., 2005). The control variables of this system are temperature, specific humidity, surface pressure, vorticity and divergence. *The resolution of the analysis grid is the same as that of AROME-WMed. Following the results of Brousseau (2012), the Incremental Analysis Update (IAU, Bloom et al., 1996) is not used for the 3DVar assimilation scheme.* Background error covariances were computed using a period characterized by convective systems in October 2010 over the northwestern Mediterranean region (Fourrié et al., 2015). Every three hours an analysis is computed by using all observations available within a $\pm$ 1 h 30 min assimilation window and a 3-h forecast is produced to provide a background for the next cycle. The assimilation system ingests a wide variety of observations from satellite, ground-based GPS, aircraft, radiosondes, drifting buoys, balloons and wind profilers, automatic land and ship weather stations, and ground-based precipitation radars of the French network ARAMIS (reflectivity and radial velocity). The purpose of this study is to assess the impact of the assimilation of RASTA data in addition to this already dense observing network.

## 2.3 RASTA data pre-processing

RASTA data are discarded between 250 m above and 250 m below the aircraft, which is the minimal measuring range of the zenith- and nadir-pointing antennas. Ground clutter is also removed. To reduce observation and representativeness errors, RASTA data are interpolated in the model vertical and horizontal resolutions. For the reflectivity measurements, this interpolation is done by taking the average value (in $mm^6$ $m^{-3}$) of all data available along the aircraft track within a box of 2.5 km length between the two half model levels surrounding each model level. From a given range from the radar, when the aircraft roll and/or pitch angles are greater than a threshold (|roll| $> 7°$ at 10 km range), some measurements might come out of the grid box of interest. Therefore, these data are removed from the interpolation. The same interpolation is done for the retrieved horizontal wind component except that a median filter is employed. Indeed, applying a median filter instead of averaging allows to reduce the influence of outliers, due to the difficulty of having high quality measurements for airborne Doppler radar (Bosart

et al., 2002).

After this pre-processing, a thinning is applied to RASTA data to decrease observation density and satisfy assumptions about observation error covariances, which are supposed to be 0 dB$^2$. It is particularly true for measurements made by different instruments, which have independent physical errors. However, this hypothesis might be no more valid if the observations are collected very close to each other by the same instrument. Applying a thinning to the observations is therefore necessary for having satisfactory assumptions about observation error covariances (Rohn et al., 2001; Liu and Rabier, 2002). Therefore, RASTA data are assimilated every 3 time steps, which is equivalent to a distance of approximately 5 km to 9 km depending on the aircraft speed. *The data are not thinned vertically because the vertical forecast error covariances are less marked than the horizontal ones (Brousseau et al., 2011) and it is thus not useful to apply any thinning in that case (Jacques and Zawadzki, 2014).*

## 3 Assimilation method

### 3.1 The 1D+3DVar assimilation method

Here, we employ the 1D+3DVar assimilation method (Caumont et al., 2010; Wattrelot et al., 2014) used operationally to assimilate ground-based precipitation radar data in AROME. This data assimilation technique allows to shift a pattern that was well simulated by the model but at a wrong location. It relies on the ability of the model to create consistent moisture and reflectivity profiles. Indeed, cloudy areas are generally associated with relative humidity close to the saturation and high reflectivity values. This method is particularly well suited for vertically pointing radar because the first step of the assimilation method is based on the differences between different vertical profiles of reflectivities. For instance, since March 2016, this assimilation method is operationally employed to assimilate vertical profiles of Dual-frequency Precipitation Radar (DPR) reflectivity data in the Japanese kilometer-scale NWP model (JMA-NHM) (Ikuta, 2016).

The first step consists of a 1D Bayesian retrieval of the best estimate of relative humidity ($RH$) profiles, named hereafter pseudo-observations ($PO$), given the observed vertical profile of reflectivity $Z_o$. For each observed column of reflectivity $Z_o$, the corresponding vertical profile of RH pseudo-observation $y_{PO}^{RH}$ is given by

$$y_{PO}^{RH} = \sum_i x_i^{RH} \frac{\exp\left(-\frac{1}{2} J_{PO}\left(x_i\right)\right)}{\sum_j \exp\left(-\frac{1}{2} J_{PO}\left(x_j\right)\right)}, \tag{1}$$

with

$$J_{PO} = \frac{1}{n_o} \frac{\displaystyle\sum_{k}^{n_o} \left(Z_{o_k} - H_z\left(x_k\right) - b_k\right)^2}{\sigma_o^2}, \tag{2}$$

where

- *subscript $i$ denotes the index of the model profile in the vicinity of the observed profile of reflectivity*

- $x_i^{RH}$ is the vertical column of relative humidity from the model background;

- $H_z\left(x_k\right)$ is the simulated reflectivity (in dBZ) at the model level $k$, given the model state $x_k$; $H_z$ being the forward operator

- $n_o$ is the number of valid observed reflectivity data in the column,

- $b_k$ is the bias correction value used at the altitude $k$ (in dB), described in subsection 3.2,

- $\sigma_o$ is the standard deviation of observation and forward operator errors (in dB).

The W-band reflectivity forward operator $H_z$ described by Borderies et al. (2018) is used to simulate the reflectivity. It is consistent with the ICE3 one-moment microphysical scheme of AROME and takes as input parameters the hydrometeor contents of the five hydrometeor species (rain, snow, graupel, cloud liquid water and pristine ice), temperature, pressure, relative humidity. The T-matrix method (Mishchenko et al., 1996) is employed to compute the single scattering properties. Following the results of Borderies et al. (2018), graupel axis ratio is set to 0.8, snow axis ratio to 0.7 and pristine ice axis ratio to 1. The forward operator returns the simulated reflectivity at each range gate from the radar and accounts for hydrometeors and water vapour attenuation.

According to Equation 1, for each observed vertical profile $Z_o$, the vertical column of RH pseudo-observation is a linear combination of the neighbouring RH profiles taken from the model background $x_i^{RH}$. The $x_i^{RH}$ neighbouring profiles are located in a 160-km-wide square centred on the aircraft location. For the AROME-WMed model, this size is sufficient to reduce the effects of spatial mismatches between model and observations (Borderies et al., 2018) and to gather a database of $x_i^{RH}$ which are consistent with the meteorological situation. In addition, the $x_i^{RH}$ profiles would become less representative with a larger size since meteorological environments can change over $\approx 100$ km.

In Equation 1, the $x_i^{RH}$ profiles are weighted by a function ($J_{PO}$) of the difference between the observed $Z_o$ and simulated $H_z\left(x_i\right)$ column of reflectivities (cf Equation 2). Thus, larger weights are given to the neighbouring columns that most closely resemble the observations. To ensure equivalent weights regardless of the number of altitude levels used for each neighbourhood profile, the square difference in Equation 2 is divided by the number of valid data over the observed column of reflectivity.

The square difference is also divided by the observation error variance $\sigma_o^2$. A small $\sigma_o$ will favour the neighbouring columns that most closely resemble the observation. However, if there is no simulated profile of reflectivity which is close enough to the observed one, there will be no retrieval since the weight tends towards a value close to 0. Hence, a small $\sigma_o$ either leads to an accurate retrieval or to no retrieval at all. On the other hand, a large $\sigma_o$ would give similar weights and smooth the neigh-

bourhood $x_i^{RH}$ profiles, regardless of which extent they resemble the observed profile of reflectivity (Caumont et al., 2010). Therefore, a sensitivity study is performed in subsection 3.3 to $\sigma_o$ values.

The Bayesian retrieval is not applied in case of clear sky, ie when all the reflectivities over the whole vertical column are below the radar sensitivity in both the simulations and the observations. However, if the simulations indicate cloud or precipi-

tation, the closest "clear-sky" profile in the vicinity of the radar is selected for the retrieval.

In the second step of the 1D+3DVar assimilation approach, the retrieved vertical profiles of relative humidity pseudo-observations $y_{PO}^{RH}$ are assimilated in the 3DVar assimilation system of AROME-WMed, like any other conventional observations.

**3.2   Bias correction**

The Bayesian retrieval can also be applied to other variables using the same weights as those from the retrieval of RH profiles (in Equation 2), for example to retrieve reflectivity pseudo-observations $y_{PO}^Z$:

$$y_{PO}^Z = \sum_i H_z(x_i) \frac{\exp\left(-\frac{1}{2} J_{PO}(x_i)\right)}{\sum_j \exp\left(-\frac{1}{2} J_{PO}(x_j)\right)}. \tag{3}$$

The reflectivity pseudo-observation $y_{PO}^Z$ can be used as an indicator to evaluate the quality of the 1D Bayesian retrieval and

to estimate the biases that can arise between observations and simulations. Indeed, in data assimilation it is necessary to have unbiased quantities and to remove these systematic errors (Janisková, 2015; Okamoto et al., 2016). These biases can arise from the observations, the ICE3 microphysical scheme and/or the forward operator formulations. Janisková (2015) showed that the biases between observations and simulations depend on the temperature and on the altitude. Since this study is focused on one specific area of interest during the same season, the bias is mainly a function of the altitude. The bias is also a function of the

error standard deviation $\sigma_o$. Indeed, while a small value of $\sigma_o$ favours the simulated columns that most closely resemble the observations, a larger value of $\sigma_o$ smooths the simulated reflectivity profiles.

Therefore, a bias $b$ was determined from the statistics between $Z_o$ and $y_{PO}^Z$ using the altitude and the error standard deviation $\sigma_o$ as predictors. Calculations were performed over all flights during the HyMeX-SOP1 every 4 time steps. The background

states of a CTRL experiment were used as a reference to simulate the reflectivity pseudo-observations and to estimate the biases. The CTRL experiment was run during a 45-day cycled period from 00 UTC 24 September 2012, which is the day when

the Falcon 20 first flew during HyMeX-SOP1, to 5 November 2012, after the last flight. It includes all the observations that are operationally assimilated (see subsection 2.2).

The bias between RASTA observations and the reflectivity pseudo-observations is depicted in Figure 2 as a function of the altitude for different values of $\sigma_o$ (see legend). Calculations were only performed if both the observation and the reflectivity pseudo-observation are above the radar sensitivity. The number of observations used for the calculations is also shown in the right panel. This number is smaller for small values of $\sigma_o$ (red curve), because it constrains the amount of retrieved profile of reflectivity pseudo-observations to only those which most closely resemble the observations.

Figure 2 shows that the bias increases with the altitude, which is consistent with the existence of model biases in cloudy areas in the ICE3 microphysical scheme of AROME-WMed (Borderies et al., 2018; Taufour et al., 2018). Figure 2 highlights the fact that, because of the smoothing effect, the bias increases with the error standard deviation $\sigma_o$. Indeed, at approximately 6 km of altitude, the bias can reach a value up to 6 dB if $\sigma_o$ equals 9 dB, and only $\approx$ 1.5 dB if $\sigma_o$ equals 2 dB.

*The effect of the bias correction is shown in Figure 3, in which Contoured Frequency by Altitude Diagram (CFAD) of the differences between the observed reflectivity and the bias-corrected reflectivity pseudo-observations are shown for a $\sigma_o$ of 2 dB. The residual bias is indicated by the black line. Figure 3 demonstrates that, after applying the bias correction in Equation 2, the residual bias is close to 0 dB except above an altitude of approximately 10 km, which is probably due to the smaller number of points used to calculate the bias correction. As explained by Janisková (2015), the use of additional predictors, such as temperature or hydrometeor contents, could lead to an improvement in the bias correction at higher altitude.*

### 3.3 Observation error within the Bayesian inversion

As explained in Equation 2, the quality of the 1D Bayesian retrieval relies on the specification of standard deviation of observation and forward operator errors $\sigma_o$ (in dB). In-flight water vapour mixing ratio measurements $r_o$ are available at flight level and can be used to estimate $\sigma_o$ and to evaluate the quality of the retrieval. These data present the advantage of being completely independent from the retrieval and they allow the evaluation of the humidity pseudo-observations which will then be assimilated in the 3DVar assimilation system of AROME-WMed.

The 1D-Bayesian retrieval is applied to the CTRL background states for error standard deviations $\sigma_o$ ranging from 0.6 dB to 9 dB. The bias correction, which has been calculated for each $\sigma_o$ in subsection 3.2, is applied in Equation 2. The retrieved pseudo-observations $r_m$ of water vapour mixing ratio at the flight level are then compared with the in-flight measurements $r_o$ over 32 flights of the HyMeX SOP1. The comparison is done as follows. First, a manual data quality control is applied to in-situ humidity observations in order to remove the poor quality measurements that can arise from instabilities or period of malfunctioning during the flights. After this quality control, it remains 24 flights out of 32. Second, water vapour mixing ratio

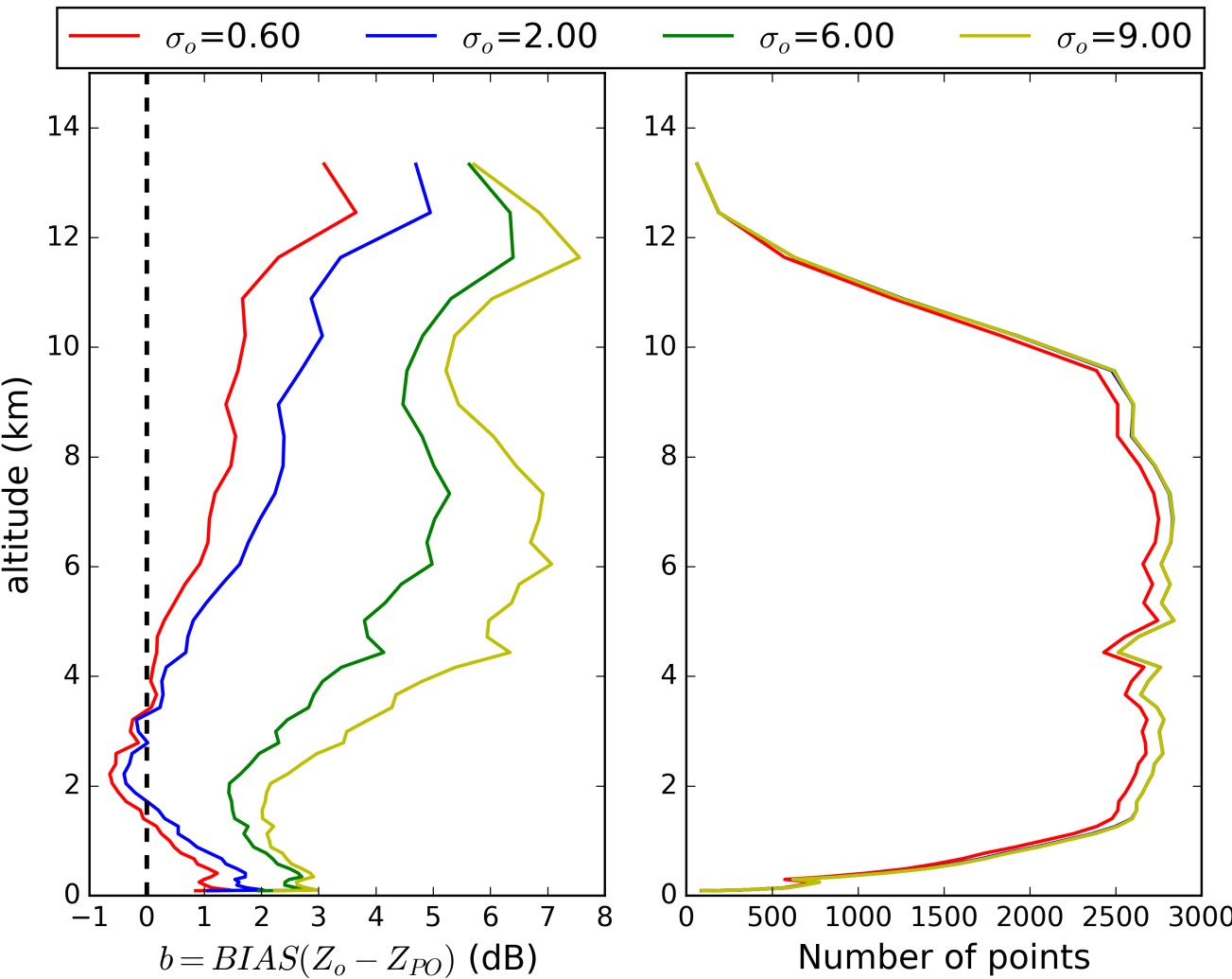

**Figure 2.** Bias (left panel) between RASTA reflectivity and the reflectivity pseudo-observation as a function of the altitude for different values of error standard deviation $\sigma_o$ (see legend box). The right panel shows the number of observations used for the calculations as a function of the altitude for the different values of error standard deviation $\sigma_o$.

measurements are averaged over 12 time steps to reduce observation noise and representation errors. Figure 4 shows the standard deviations (right panel) and biases (left panel) between the observed in-flight water vapour mixing ratio and the retrieved ones (red curve) as a function of the error standard deviations $\sigma_o$. The standard deviations between in-flight measurements and the background water vapour mixing ratio are also represented by the black data points.

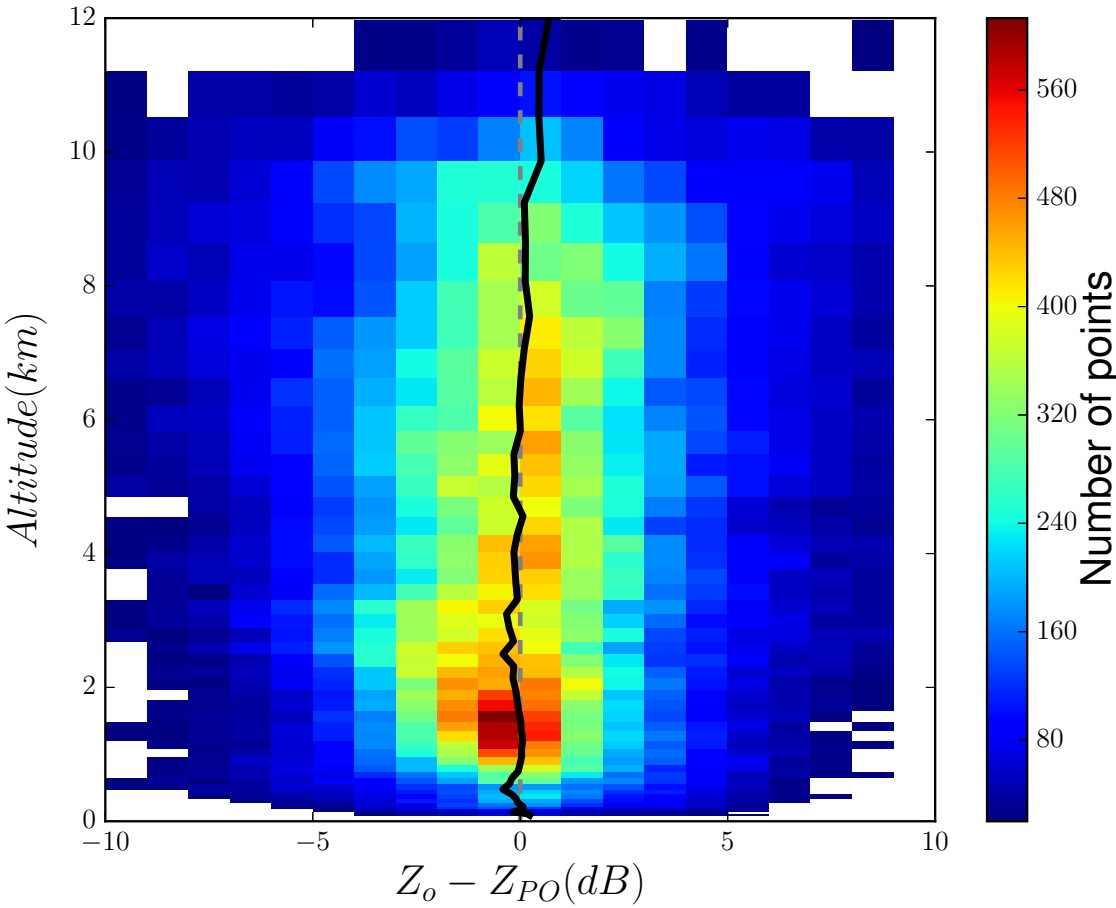

**Figure 3.** *Contoured Frequency by Altitude Diagram (CFAD) of the differences between the observed reflectivity and the bias-corrected reflectivity pseudo-observations. The residual bias after applying the bias correction is indicated by the black line.*

The standard deviation values in Figure 4 demonstrate that the retrieved water vapour mixing ratios are always in better agreement with the in-flight measurements, compared to the background state. This improvement highlights the ability of the 1D Bayesian method to retrieve humidity fields that are closer to independent observations. The variation of the standard deviation indicates the existence of an optimal value of $\sigma_o$ of approximately 2 dB. Indeed, below 2 dB, the standard deviation increases with decreasing $\sigma_o$. This is due to the tendency of the retrieval to be more selective for small values of $\sigma_o$, which results in using the background state instead of applying the retrieval. On the contrary, above 2 dB, the standard deviation increases with $\sigma_o$. Indeed, a large $\sigma_o$ increases the number of successful inversions, but smooths them to produce the resulting

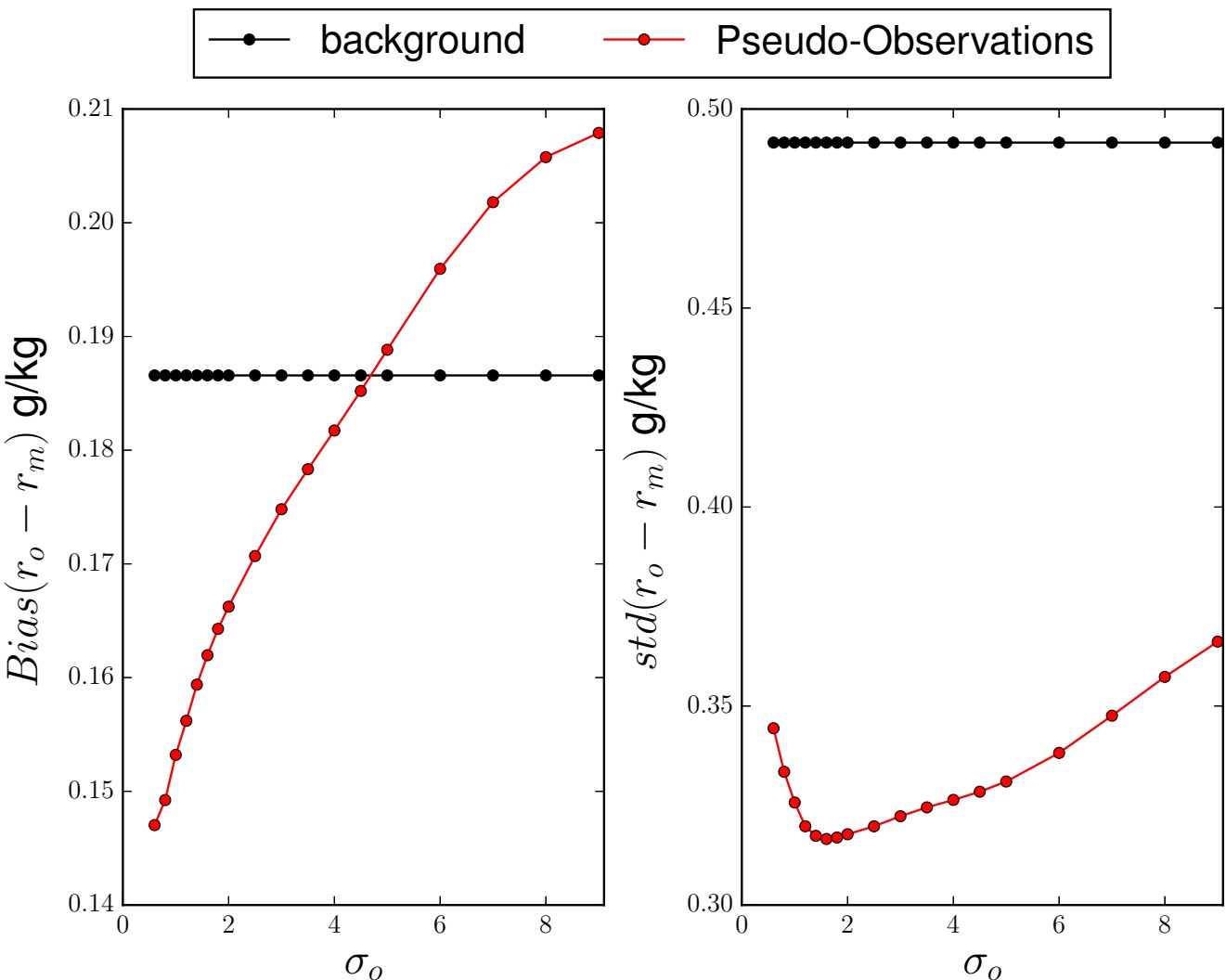

**Figure 4.** Standard deviations (right panel, in g/kg) and biases (left panel, in g/kg) *of the water vapour mixing ratio differences between in-flight measurements $r_q$ and the retrieved ones $r_m$ (red) as a function of the error standard deviations $\sigma_o$ (in dB).* The standard deviations and biases before applying the 1D-Bayesian retrieval are represented by the black data points.

humidity pseudo-observations. Finally, it should be noted that the bias is also improved with a $\sigma_o$ of 2 dB (left panel). Hence, we decided to use an error standard deviation $\sigma_o$ of 2 dB for the rest of this study.

## 4 Data assimilation experiments

To assess the potential of RASTA data to improve analyses and forecasts of heavy precipitation events, a total of 4 experiments is conducted. The CTRL experiment includes all the observations that are operationally assimilated (see subsection 2.2). Three additional RASTA experimental designs (Z, V, ZV) share the same configuration as CTRL, except that they also include the assimilation of RASTA data. The reflectivity observed by RASTA is assimilated alone in the Z experiment, and jointly with RASTA horizontal wind components in the ZV experiment. The 1D+3DVar assimilation method described in subsection 3.1 is employed to assimilate RASTA reflectivity observations in the Z and ZV experiments. In addition, the V experiment includes the assimilation of RASTA wind data alone. As explained by Borderies et al. (in review), the assimilation of RASTA wind data is straightforward and does not require the use of a radial wind observation operator. Indeed, the Doppler multi-beam antenna system of RASTA allows the retrieval of the horizontal wind components $(u, v)$, which are directly linked to two control variables of the AROME model (vorticity and divergence).

RASTA data are not measured simultaneously, but over the flight leg. Consequently, at each assimilation cycle, the 3DVar assimilation system of AROME-WMed ingests all the RH pseudo-observations and/or RASTA wind data available during a 2-hour assimilation window centred on the assimilation time $T$, as if they were valid at the time $T$. A larger assimilation window increases the number of observations and results in larger coverage. However, since RH pseudo-observations vary with convective systems, which can evolve quickly in time, a larger assimilation window would result in assimilating data that are no longer valid at the current assimilation time $T$. Besides, Borderies et al. (in review) conducted a sensitivity study to the length of the assimilation window, by assimilating airborne Doppler wind radar data in the 3DVar assimilation system of AROME-WMed. Results indicated that, even though the best scores were reached with a 1 hour assimilation window, a slight positive improvement of the 6-hour precipitation forecasts was also evidenced with a 2-hour assimilation window. Therefore, a 2-hour assimilation window is a good compromise to assimilate a larger number of observations, which are nearly valid at the assimilation time, without adding any detrimental observation in the assimilation system. Hence, the length of the assimilation window has been set to 2 hours in this study.

The observation errors for the RH pseudo-observations $y_{PO}^{RH}$ and RASTA wind data are the same as the one used for the radiosonde measurements. It is set to 12% for the RH pseudo-observations. RASTA wind observation error increases with the altitude from approximately 1.8 m s$^{-1}$ at 900 hPa to approximately 2.5 m s$^{-1}$ at 200 hPa. Finally, in addition to the pre-processing described in subsection 2.3, a quality control is also performed prior to the assimilation: observations with innovation (Observations - Background) greater than a threshold are rejected. This threshold depends on both the observation and background errors. It has a constant value of approximately 55% for the RH pseudo–observations. It increases with the altitude for RASTA wind data because the error standard deviation is a function of the altitude (approximately 25 m s$^{-1}$ at the maximum).

The four different experiments are first conducted for a heavy precipitation event which occurred during the Intensive Observing Period 7a (IOP7a) over South-Eastern France on 26 September 2012. During this case study, RASTA data were collected during Flight 15 between 06:10 UTC and 09:45 UTC (red line in Figure 1). Therefore, RASTA data are assimilated for the first time at the 06:00 UTC analysis. The different experiments are named CTRL$^{IOP7}$, Z$^{IOP7}$, V$^{IOP7}$ and ZV$^{IOP7}$. They share the same background field to compute the 06:00 UTC analysis. They start at 00 UTC 26 September 2012 and end at 12 UTC 26 September 2012. Next, in order to study the impact of the assimilation of RASTA data in various conditions during the whole HyMeX-SOP1, the four experiments are run for the 32 analysis cases in which RASTA data are available. For this configuration, the CTRL experiment is the same as the one used in subsection 3.2, which was run during a 45-day cycled period from 00 UTC 24 September 2012 to 5 November 2012. For the sake of simplicity, the CTRL experiment is named CTRL$^{SOP1}$. The three RASTA experiments are respectively named Z$^{SOP1}$, V$^{SOP1}$ and ZV$^{SOP1}$. In order to disentangle the cycling effect from the impact of the assimilation of RASTA data on the analyses, the Z$^{SOP1}$, V$^{SOP1}$ and ZV$^{SOP1}$ experiments are not cycled and share the same background fields as the CTRL$^{SOP1}$ experiment ones.

## 5 Impact on the IOP7a case study

To assess the potential of RASTA observations to improve short-term forecasts, focus is first made on one of the most significant precipitation events which occurred during IOP7a. More than 100 mm of rain were observed between 00:00 UTC on 26 September and 00:00 UTC on 27 September in the area delimited by the red box in Figure 1 (Hally et al., 2014). As mentioned in section 4, RASTA data are assimilated for the first time at the 06:00 UTC analysis in the Z$^{IOP7}$, V$^{IOP7}$ and ZV$^{IOP7}$ experiments. Most of these data are located upwind of where the rainfall event took place later in the morning at approximately 08:00 UTC. Such a configuration is required to evaluate the impact of the assimilation of RASTA data to improve heavy precipitation events.

### 5.1 1D Bayesian retrieval

As explained in subsection 3.1, the first step to assimilate the reflectivity consists of a 1D-Bayesian retrieval of Relative Humidity (RH) pseudo-observation profiles, given the vertical profile of reflectivity observed by RASTA. Since no direct RH observations are available with such a high vertical resolution as the one of RASTA data, the method is validated by comparing the reflectivity pseudo-observations $y^Z_{PO}$ with RASTA $Z_o$ observations. Figure 5 shows RASTA observations (interpolated on the vertical grid model, A), the simulated profile of reflectivities from the background (B) and the retrieved reflectivity pseudo-observations (C). The differences between the RH pseudo-observations and the background RH profiles are also shown in the bottom panel (D). Differences are displayed in red (blue) if RH pseudo-observations are larger (smaller) than the background.

Figure 5 highlights the capability of the 1D-Bayesian method to retrieve profiles which are in better agreement with the observations than the background. For example, at approximately 06:30 UTC, the observation profiles indicate clouds below an altitude of 6 km, as opposed to the simulated profiles from the background which only indicate clear sky profiles. This

has been rectified in the reflectivity pseudo-observation profiles, and in the corresponding RH pseudo-observations profiles. Indeed, the RH pseudo-observations values are larger than the background RH values (red values in D), and are thus more representative of the presence of a cloud. Inversely, at approximately 06:25 UTC, the Bayesian retrieval has been able to remove the low level clouds present in the background, and to add clouds above an altitude of about 4 km. Between 06:50 UTC and

07:00 UTC, the reflectivity pseudo-observations are also in much better agreement with the observations than the background. The corresponding RH pseudo-observations values are also larger than the background, which is consistent with the fact that larger RH values are usually associated with larger reflectivity values. Hence, Figure 5 demonstrates the ability of the Bayesian retrieval to pick-up vertical profiles in the neighbourhood which are more consistent with the observations. Indeed, this retrieval successfully dried areas associated with low reflectivity values, and moistened areas associated with high reflectivity values.

These retrieved RH pseudo-observation profiles are then assimilated in the 3DVar assimilation system of AROME-WMed in the $Z^{IOP7}$ and $ZV^{IOP7}$ experiments.

## 5.2  Impact on analyses

Figure 6 shows (from the top to the bottom), the relative humidity for the pseudo-observations, the $CTRL^{IOP7}$, the $Z^{IOP7}$, the $V^{IOP7}$ and the $ZV^{IOP7}$ analyses. Similarly, Figure 7 represents the wind speed for the observations and the different experiments.

The four different analyses were computed using the same background state.

As shown in Figures 6 and 7 (top panel), the number of RH pseudo-observations which have been assimilated is larger than the number of RASTA wind data. Indeed, contrary to RASTA wind data, the reflectivity is also assimilated in case of clear sky. Besides, airborne Doppler velocity measurements are contaminated by the aircraft motion (roll/pitch/drift angles, ground

speed, etc.). Therefore, because of the difficulty to have high quality measurements (Bosart et al., 2002), RASTA wind data have been more frequently rejected (Cf between 06:42 UTC and 06:50 UTC in Figure 7). In addition, contrary to the W-band reflectivity measurements, RASTA horizontal wind components are obtained through a retrieval, which might also explain the smaller number of assimilated horizontal wind data. Finally, since RH pseudo-observations are assimilated in the same way as radiosonde observations are (Cf section 4), they are rejected above an altitude of approximately 9 km because the values are

very small.

Compared to the RH pseudo-observations in Figure 6, RH is overestimated in the $CTRL^{IOP7}$ and in the $V^{IOP7}$ analyses. Except at approximately 8 km of altitude, the RH profiles are much more similar to the RH pseudo-observations in the $Z^{IOP7}$ and $ZV^{IOP7}$ analyses. Conversely, in Figure 7, the $V^{IOP7}$ and $ZV^{IOP7}$ analyses are in much better agreement with RASTA wind

observations compared to the $CTRL^{IOP7}$ and the $Z^{IOP7}$ analyses. Figure 6 shows that the $V^{IOP7}$ analysis is very similar to the $CTRL^{IOP7}$ one in terms of humidity. Similarly, in Figure 7, the $Z^{IOP7}$ analysis is very similar to the $CTRL^{IOP7}$ one in terms of wind speed. Therefore, the assimilation of RASTA wind data (resp. RH pseudo-observations) does not impact the humidity (resp. wind) field in the analysis, probably because wind and humidity are not highly correlated in the assimilation process

through the background error covariances. However, the assimilation of the RH pseudo-observations jointly with RASTA wind data (ZV$^{IOP7}$ experiment) results in a positive impact in terms of both the wind and the humidity fields.

### 5.3   Impact on rainfall forecasts

Figure 8 shows the 12-hour accumulated rainfall between 06:00 UTC and 18:00 UTC 26 September 2012 (IOP7a) for radar
5    observations, the CTRL$^{IOP7}$, the Z$^{IOP7}$, the V$^{IOP7}$ and the ZV$^{IOP7}$ experiments.

First, the predicted rainfall pattern is well reproduced in the four different experiments. As shown by Borderies et al. (in review), the maximum rainfall accumulation is overestimated in the CTRL$^{IOP7}$ experiment ($\approx$ 142 mm versus 93 mm in the radar observations), but is better reproduced in the Z$^{IOP7}$ experiment (130 mm). In addition, the assimilation of RH pseudo-observations jointly with RASTA wind data in the ZV$^{IOP7}$ experiment also results in a decrease (133.5 mm) of the predicted
10    maximum rainfall accumulation. Finally, the experiment in which RASTA wind data are assimilated alone in the V$^{IOP7}$ leads to the better agreement with the radar observations. Indeed, the maximum rainfall forecast accumulation has been reduced to only 118 mm.

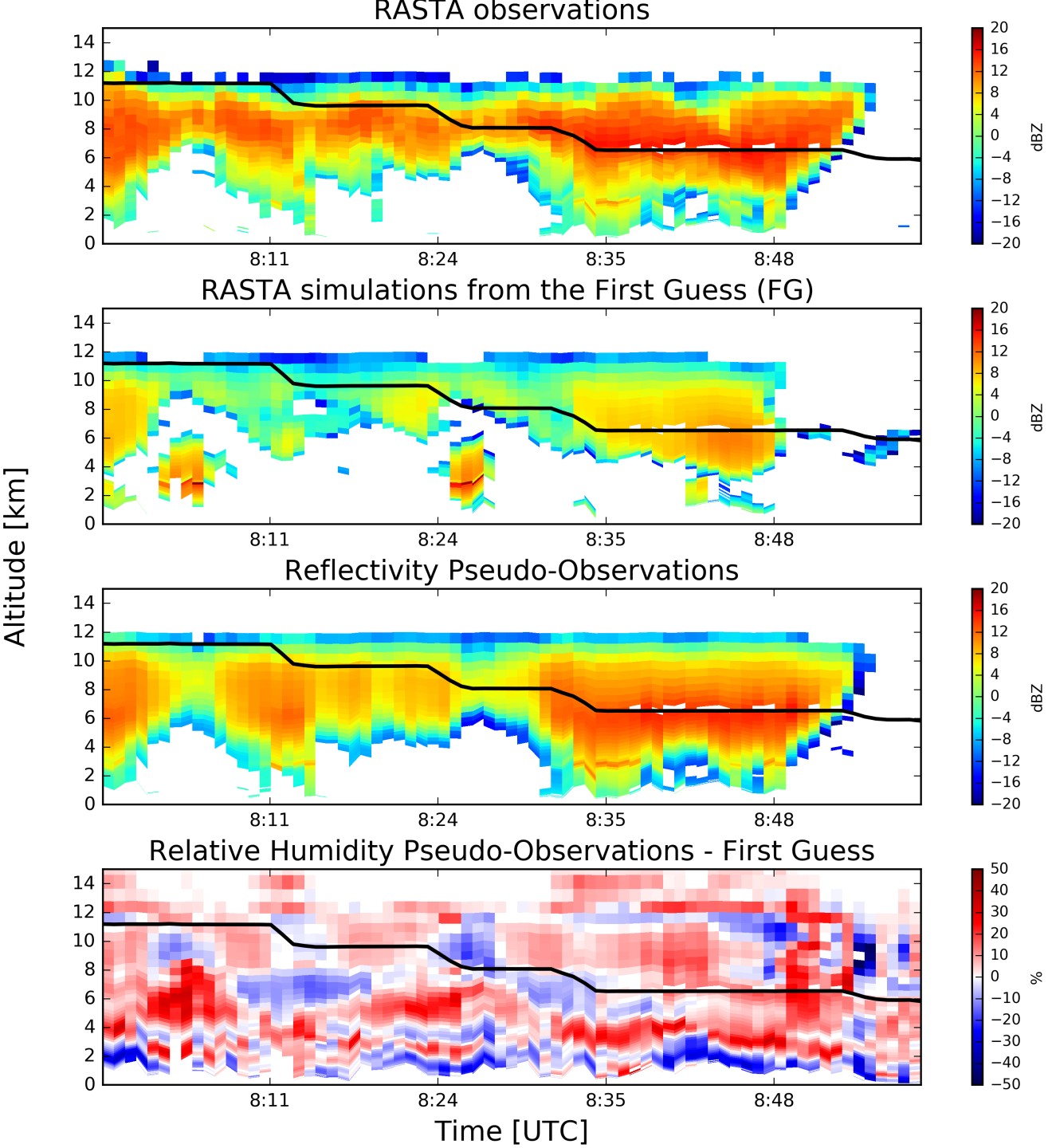

**Figure 5.** Time-height cross section of the reflectivity observed by Rasta (A), simulated from the background (B), and pseudo-observations (C). The differences between the Relative Humidity pseudo-observations and the relative humidity from the background state are shown in Figure D. Aircraft altitude is indicated by the black line.

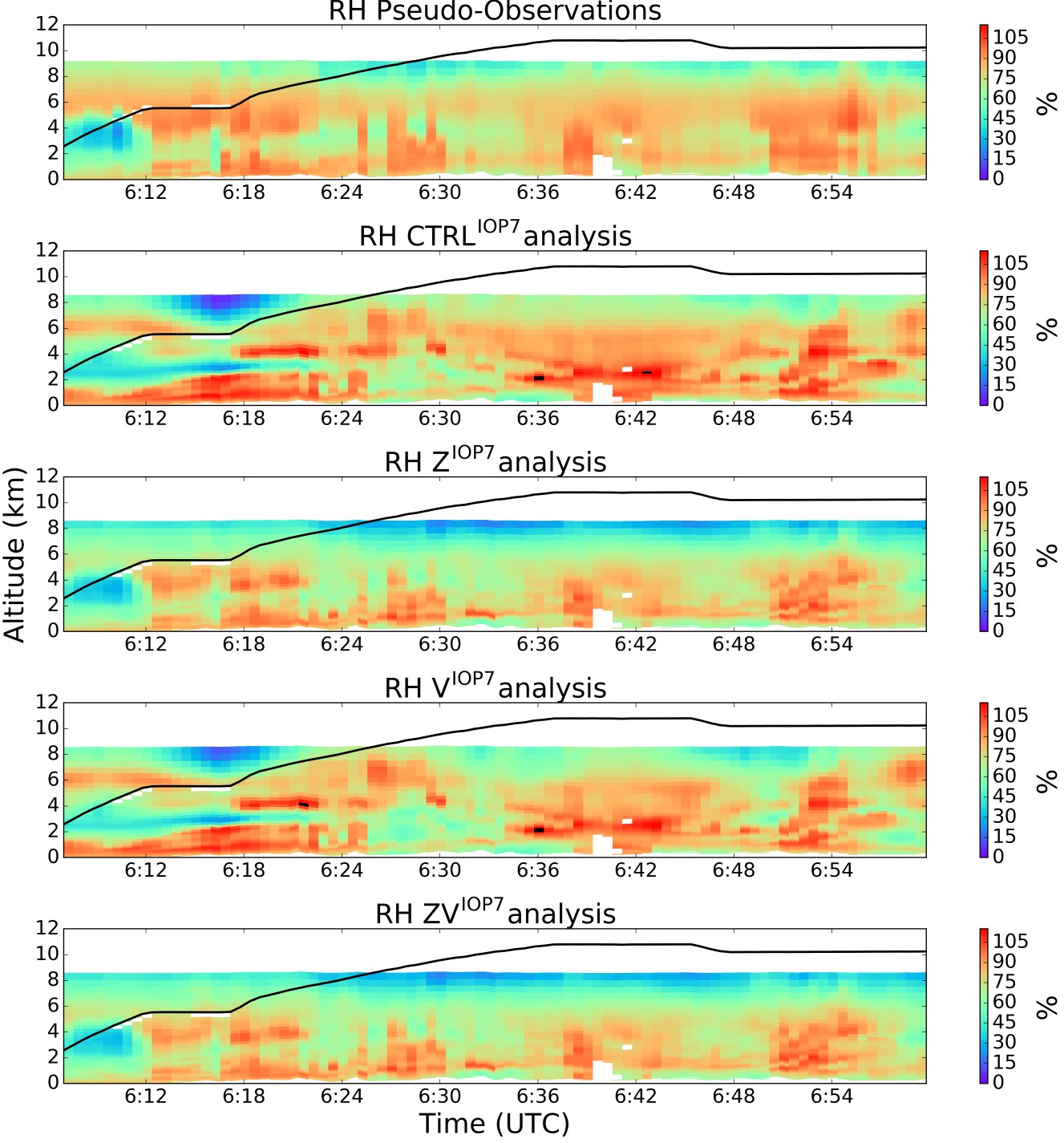

**Figure 6.** Relative Humidity (RH, in%) for (from the top to the bottom) the pseudo-observations, the CTRL[IOP7], the Z[IOP7], the V[IOP7] and the ZV[IOP7] experiments.

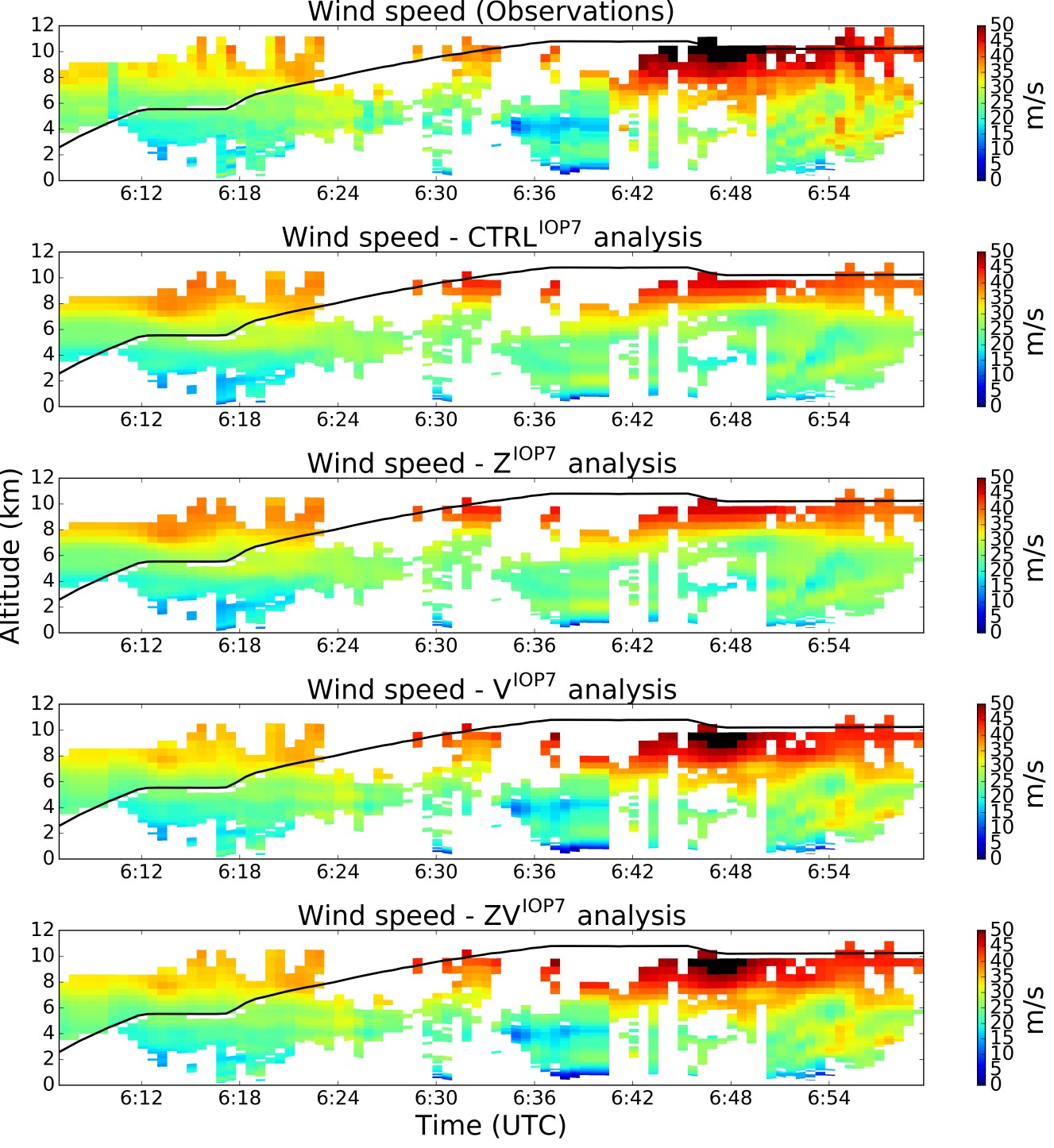

**Figure 7.** Wind speed (m/s) for (from the top to the bottom) the observations, the CTRL[IOP7], the Z[IOP7], the V[IOP7] and the ZV[IOP7] experiments.

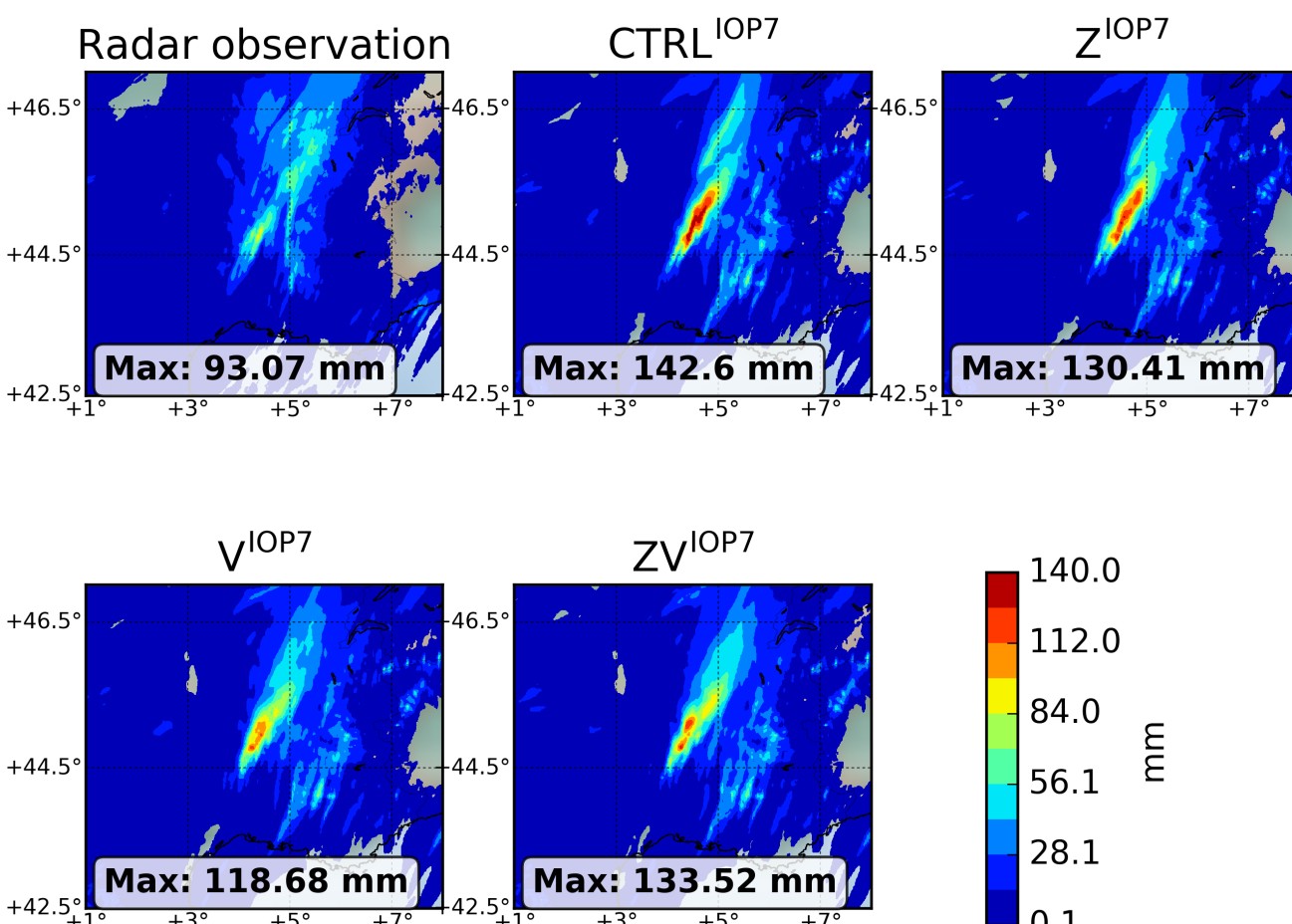

**Figure 8.** 12-hour accumulated rainfall between 06:00 UTC and 18:00 UTC 26 September 2012 (IOP7a) for radar observations, the CTRL$^{IOP7}$, the Z$^{IOP7}$, the V$^{IOP7}$ and the ZV$^{IOP7}$ experiments.

## 6  Results on the HyMeX SOP1

The impact of the assimilation of RASTA data is now assessed over the 32 cases in which RASTA data were assimilated during the HyMeX-SOP1. In order to use the same background fields, we use the $Z^{SOP1}$, $V^{SOP1}$ and $ZV^{SOP1}$ experiments, which are not cycled. An exergy distance-based approach (Marquet et al., 2019) is first employed to measure the relative impact of the assimilation of RASTA observations on the analysis and forecast fields. Then, the added value of the assimilation of RASTA data on the analyses is evidenced by using independent humidity measurements. Finally, the subsequent forecasts are validated against rain-gauge measurements.

### 6.1  Impact study using an exergy distance-based approach

The moist-air available-enthalpy (exergy) distance (Marquet et al., 2019) is first briefly described, and then calculated to measure the relative impact of the assimilation of RASTA data on analyses and short-term forecasts.

#### 6.1.1  The moist-air available-enthalpy (exergy) distance

Traditionally, the impact of the assimilation of a new observation type and its synergistic effect with other observations are assessed through verification scores of a long data-denial assimilation experiment (Storto and Randriamampianina, 2010). These approaches are very expensive from a numerical point of view. The new type of observation needs to be assimilated in a large number of analysis cases, which can not be affordable for airborne radar measurements since the availability of the new observation depends on the aircraft flights. By contrast, energy-based approaches (Ehrendorfer et al., 1999; Marquet et al., 2019) are cost effective methods for evaluating the impact of the assimilation of a new observing system in a NWP model. The idea is to combine thermodynamic variables of the atmosphere into a model space-based measure (Storto and Randriamampianina, 2010), which avoids the use of long data-denial experiments and adjoint-based methods, that rely on strong linearity assumptions which are not valid at the convective scale. These approaches provide a measure of the relative impact of the observations on the analysis and forecast fields. For example, Storto and Randriamampianina (2010) employed the Moist Total Energy Norm (Ehrendorfer et al., 1999, MTEN) to evaluate the loss of quality in the forecasts when an observation type is not assimilated. A similar methodology was employed by Fabry and Sun (2010) to characterize model errors in winds, temperature, humidity, and precipitation.

Based on results of (Marquet, 1993), Marquet et al. (2019) defined a moist-air available-enthalpy (exergy) distance, which provides a more general and comprehensive metric between a perturbed thermodynamic state (here: the RASTA experiments), and a reference one (here: the CTRL experiment). It is defined by the integration over the 2D domain of the sum of four quadratic terms in horizontal wind components $U, V$ ($N_s$), temperature $T$ ($N_T$), surface pressure $p_s$ ($N_p$) and water vapour

mixing ratio $r_v$ ($N_v$). The four contribution terms of the exergy distance are then given by

$$N_T = \int\limits_D \frac{C_{pd}T_r}{2} \frac{\left(T^{CTRL} - T^i\right)^2}{\overline{T^{CTRL}}^2} dD, \tag{4}$$

$$N_P = \int\limits_D \frac{R_d T_r}{2} \frac{\left(p_s^{CTRL} - p_s^i\right)^2}{\overline{p_s^{CTRL}}^2} dD, \tag{5}$$

$$N_v = \int\limits_D \frac{R_v T_r}{2} \frac{\left(r_v^{CTRL} - r_v^i\right)^2}{\overline{r_v^{CTRL}}} dD, \tag{6}$$

$$N_s = \int\limits_D \frac{\left(U^{CTRL} - U^i\right)^2 + \left(V^{CTRL} - V^i\right)^2}{2} dD, \tag{7}$$

where the superscript $i$ denotes the RASTA experiments (Z$^{SOP1}$, V$^{SOP1}$ or ZV$^{SOP1}$), $C_{pd}$ is the specific heat of dry air, $R_d$ is the dry air constant, $R_v$ is the water vapour gas constant, and $T_r$ is the reference temperature (taken to 300 K). The total exergy distance is then given by the sum of the four quadratic terms $N_T$, $N_p$, $N_v$ and $N_s$.

In equations 4, 5 and 6, the contribution terms of the exergy distance are divided by the weighting factors $\overline{T^{CTRL}}$, $\overline{p_s^{CTRL}}$ and $\overline{r_v^{CTRL}}$, which correspond to the average values over the 2D domain of $T^{CTRL}$, $p_s^{CTRL}$ and $r_v^{CTRL}$, respectively. Hence, as defined by Ehrendorfer et al. (1999), the weighting factors $\overline{T^{CTRL}}$ and $\overline{r_v^{CTRL}}$ are a function of the altitude, where an arbitrary factor "$\epsilon$" was introduced however with unknown values between 0.1 and 10. This arbitrariness is removed by Marquet et al. (2019) where $\overline{r_v^{CTRL}}$ varies significantly with height, since the water vapour mixing ratio decreases by 3 orders of magnitude between the surface and the stratosphere. Therefore, moisture analysis and forecast impacts between the different atmospheric levels are fully taken into account through the use of these altitude-dependent weighting factors. Hence, the use of the exergy distance is expected to more fairly rank the different observing systems through the use of more balanced contributions between wind, temperature and water vapour.

In this study, the four different contribution terms of the exergy distance will be studied independently in order to evaluate the respective impact of the assimilation of the RH pseudo-observations and/or RASTA wind components on temperature (Equation 4), surface pressure (Equation 5), water vapour mixing ratio (Equation 6) and wind (Equation 7) fields.

### 6.1.2 Impact on analyses

The temperature $N_T$, surface pressure $N_p$, humidity $N_v$ and kinetic $N_s$ contribution terms of the exergy distance are calculated over the domain defined by the black box in Figure 1. Figure 9 represents $N_T$ (A), $N_p$ (B), $N_v$ (C) and $N_s$ (D) as a function of the altitude for the Z$^{SOP1}$ (red curve), V$^{SOP1}$ (blue curve) and ZV$^{SOP1}$ (black curve) analyses. The different contribution terms are averaged over the 32 analyses in which RASTA data have been assimilated.

First, Figure 9 demonstrates that the assimilation of RH pseudo-observations and/or RASTA wind data has a small impact on the temperature $N_T$ (A) and surface pressure $N_p$ (B) contribution terms of the exergy distance. Indeed, even though there is a correlation between the different variables through the background error covariance matrix (Fabry and Sun, 2010), there

is a larger impact on the contribution terms ($N_v$ and/or $N_s$) that are associated to the variables (wind and/or humidity) directly linked to the assimilated observations. On the analyses, the experiment which has the smallest impact on $N_T$ (A) and $N_p$ (B) is the V$^{SOP1}$ experiment (blue curve), followed by the Z$^{SOP1}$ experiment. However, this rank order is reversed after only one-hour forecast (not shown). The larger impact on $N_T$ and $N_p$ is obtained if RH pseudo-observations are assimilated jointly with

RASTA wind data (ZV$^{SOP1}$ experiment, black curve).

As expected, since RH pseudo-observations are linked to the humidity fields, the impact of the assimilation of RH pseudo-observations (Z$^{SOP1}$) is larger on $N_v$ than on the other contribution terms. Similarly, since RASTA wind observations are linked to the horizontal wind components, their assimilation (V$^{SOP1}$) result in a larger impact on $N_s$. Next, the assimilation of RH pseudo-observations (resp. RASTA wind data) does not impact significantly $N_s$ (resp. $N_v$). Therefore, humidity and horizontal

wind data do not seem to be highly correlated with one another, which is consistent with the results of subsection 5.2. A larger impact on both the $N_v$ and $N_s$ contribution terms is obtained if RH pseudo-observations and RASTA wind data are assimilated jointly (ZV$^{SOP1}$, black curve), along with a larger impact on all the contribution terms. Consequently, this result indicates that the joint assimilation is required to have an impact on both the wind and humidity fields in the analyses.

### 6.1.3 Impact on short-term forecasts

Equations 4 and 6 are now integrated over the two dimensional domain and the vertical levels for different forecast terms. Results are only shown for the kinetic ($N_s$) and humidity ($N_v$) contribution terms of the exergy distances because the major differences have mainly been evidenced on these two terms (see Figure 9). Figure 10 represents $N_s$ (C and D) and $N_v$ (A and B) as a function of the forecast term over land (left panels) and over sea (right panels) for the Z$^{SOP1}$ (red curve), V$^{SOP1}$ (blue curve) and ZV$^{SOP1}$ (black curve) experiments.

Generally, Figure 9 shows that the impact is larger over sea (right panels) than over land (left panels). Indeed, ground-based precipitation radar data (reflectivity and Doppler velocity) are also assimilated over land. Therefore, there is a lack of wind and humidity observations over sea, which is partly compensated by the assimilation of RASTA data. This is particularly evidenced for the humidity contribution term $N_v$ of the exergy distance (A and B panels). However, after 2-hour forecast term, the impact of the V$^{SOP1}$ and ZV$^{SOP1}$ experiments on $N_s$ is of the same order of magnitude over land and over sea.

Except at the analysis time on the humidity contribution term $N_v$, the impact of the assimilation of RH pseudo-observations (Z$^{SOP1}$ experiment) is always smaller than the impact of RASTA wind data (V$^{SOP1}$ experiment). This can be attributed to the fact that the forecast system seems to have a short memory of RH pseudo-observations, which is consistent with the findings of Storto and Tveter (2009). The impact of the assimilation of RASTA wind data has a larger impact on the humidity forecasts, probably by adjusting large structures, and by modifying in return the frontal and/or convective features. In addition, the impact

is always larger when RH pseudo-observations are assimilated jointly with RASTA wind data (ZV$^{SOP1}$ experiment, black curve). This result was expected because the number of assimilated observations has been increased in the ZV$^{SOP1}$ experiment. Besides, the ZV$^{SOP1}$ experiment seems to take the benefits (or disadvantages) of both the Z$^{SOP1}$ and the V$^{SOP1}$ experiments. The small impact of the Z$^{SOP1}$ experiment seems to indicate that it is pointless to assimilate RH pseudo-observations without modifying in a consistent way the wind field. Finally, the ZV$^{SOP1}$ experiment leads to a larger impact on the kinetic contribution

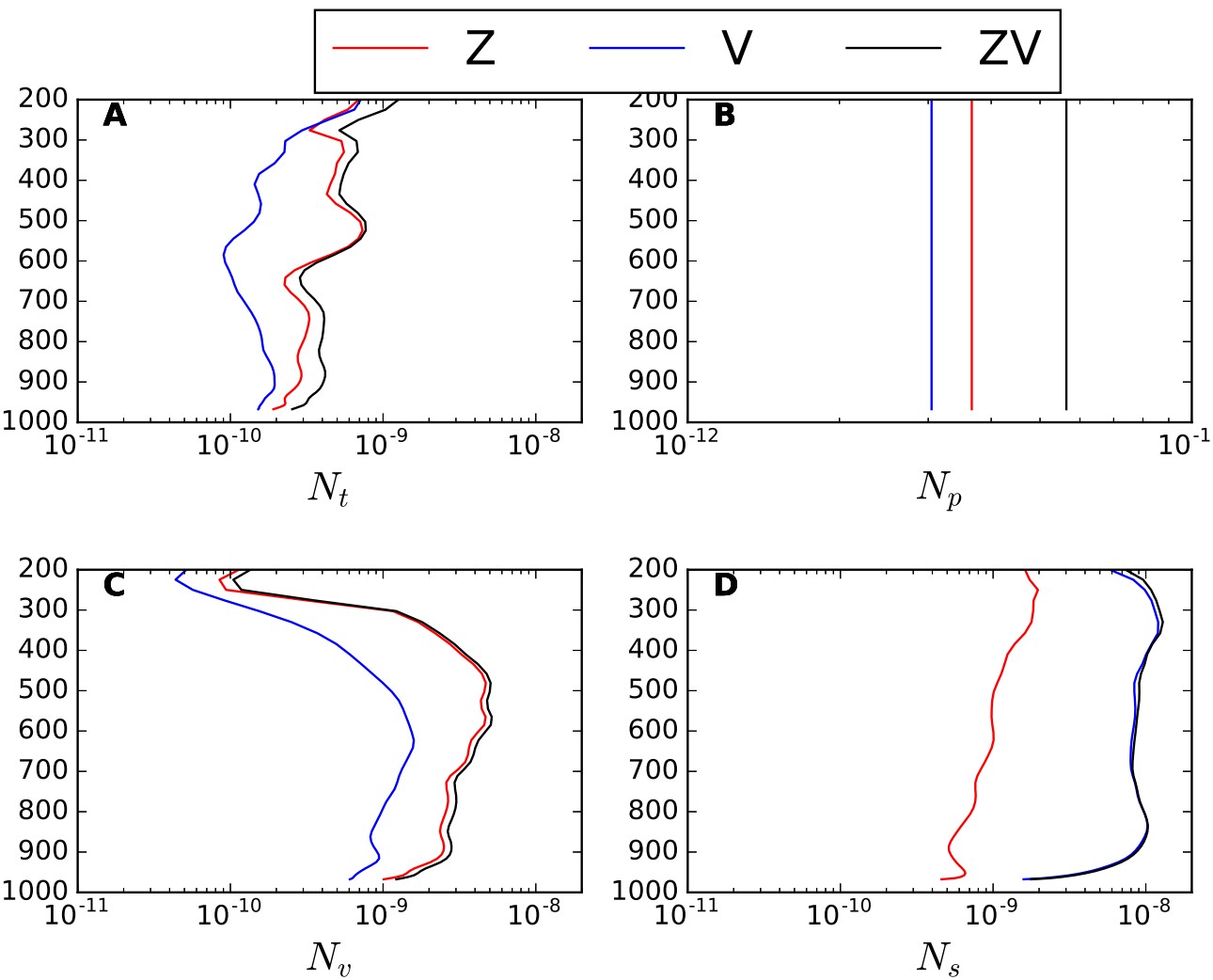

**Figure 9.** Temperature $N_T$ (A), surface pressure $N_p$ (B), humidity $N_v$ (C) and kinetic $N_s$ (D) contribution terms of the exergy distance as a function of the altitude for the $Z^{SOP1}$ (red curve), $V^{SOP1}$ (blue curve) and $ZV^{SOP1}$ (black curve) experiments.

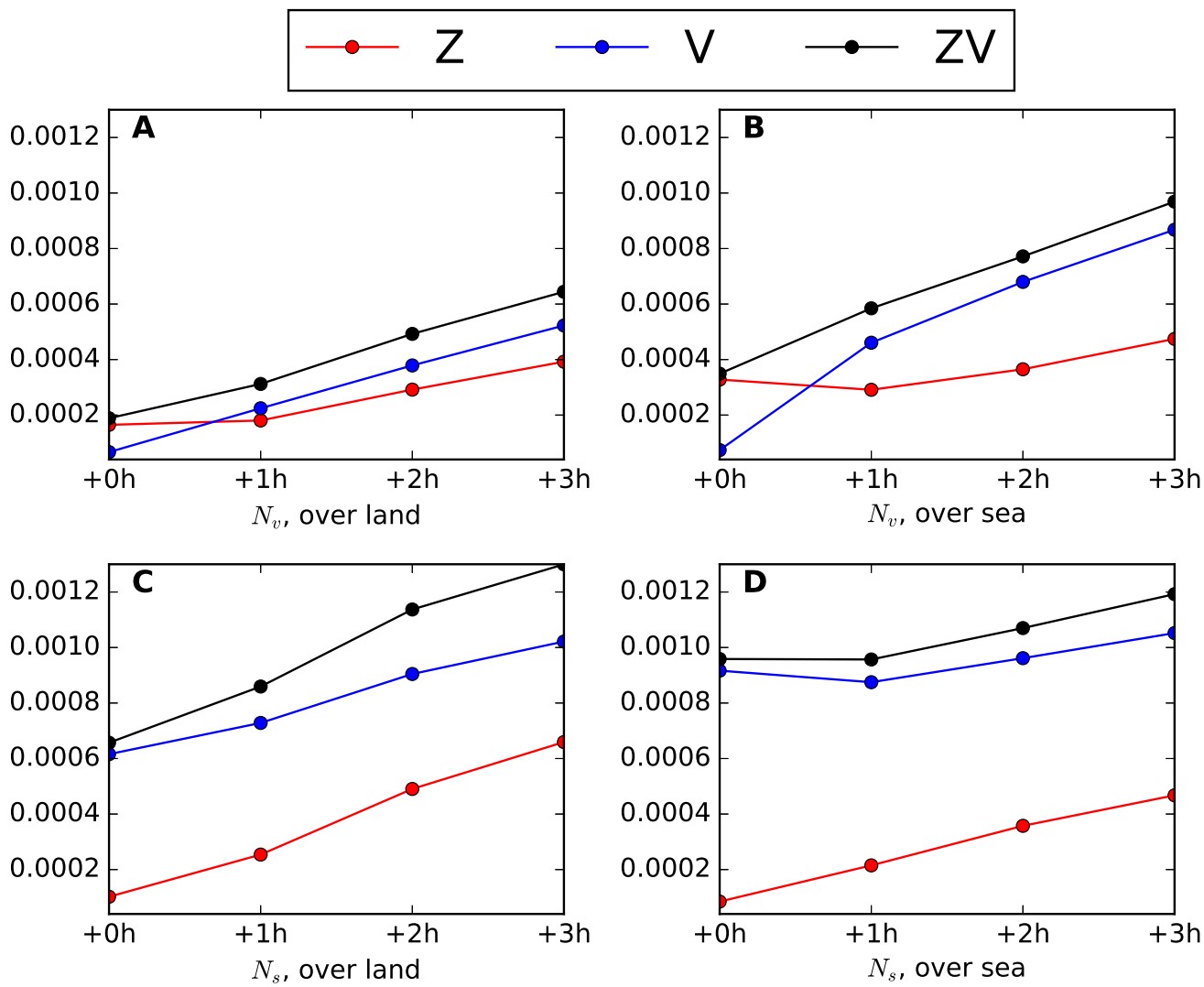

**Figure 10.** Temperature ($N_s$) (C and D) and humidity ($N_v$) (A and B) contribution terms of the exergy distances as a function of the forecast term over land (left panels) and over sea (right panels) for the Z[SOP1] (red curve), V[SOP1] (blue curve) and ZV[SOP1] (black curve) experiments.

term $N_s$ than on the humidity contribution term $N_v$. This is can be explained by the fact that the V$^{\text{SOP1}}$ experiment has more impact on $N_s$ than the Z$^{\text{SOP1}}$ experiment has on $N_v$.

To conclude, the relative impact of the assimilation of RASTA data on the analysis and forecasts fields has been evidenced using the exergy distance. This impact study highlighted that RH pseudo-observations have a modest impact on the analyses on the humidity field, which vanishes soon as the forecast term increases compared to the experiment in which RASTA wind data are assimilated alone. The impact on the subsequent forecasts is more important if both data are assimilated jointly. The benefit brought by this impact will be evaluated in the next sections.

## 6.2   Analyses evaluation: comparisons against in situ measurements

The aim of this section is to assess the added value of the assimilation of RASTA data on the analyses. The evaluation is not shown against other conventional assimilated observations, because, as expected, the fit to observations is always better in CTRL$^{\text{SOP1}}$ than in the RASTA experimental analyses. However, in-flight humidity measurements at flight level are not assimilated in any of the experiments, and are used as independent observations to assess the impact of the assimilation of RASTA data on the humidity analyses. As explained in subsection 3.3, poor quality measurements are removed for the comparisons. Hence, after the manual quality control, it only remains 24 analysis cases. Figure 11 shows the standard deviation between humidity mixing ratio measurements and the analysed ones for the different experiments (Z$^{\text{SOP1}}$ , V$^{\text{SOP1}}$ and ZV$^{\text{SOP1}}$) during the 24 analysis cases. The standard deviation between the measurements and the water vapour mixing ratios from the background state is also represented by the black data points, which is a constant value because the same background states are used in all the different experiments.

First, it should be noted that the analysed water vapour mixing ratios are always in better agreement with the observations compared to the background field, which is quite reassuring. Next, the standard deviation is slightly larger for the V$^{\text{SOP1}}$ than for the CTRL$^{\text{SOP1}}$ experiment. Hence, the assimilation of RASTA wind data alone (V$^{\text{SOP1}}$) does not improve the analysis in terms of humidity, which was expected because RASTA wind data are only slightly related to humidity, so it is likely that the humidity analysis field moves away from humidity observations. The experiment that reduces the most the standard deviation is the Z$^{\text{SOP1}}$ experiment, which indicates that the assimilation of RH pseudo-observations alone impacts positively the analysis in terms of humidity.

Even though slightly less pronounced, the assimilation of RH pseudo-observations jointly with RASTA wind data (ZV$^{\text{SOP1}}$ experiment) also leads to an improvement of the analysed humidity field. The respective impacts of the Z$^{\text{SOP1}}$ and V$^{\text{SOP1}}$ experiments are both present in the ZV$^{\text{SOP1}}$ experiment. Therefore, since the standard deviation is slightly larger for the V$^{\text{SOP1}}$ experiment, it seems logical that the standard deviation in Figure 11 is larger for the ZV$^{\text{SOP1}}$ experiment than for the Z$^{\text{SOP1}}$. In addition, it has been demonstrated in Figure 9 that the humidity field in the analysis is more impacted by the assimilation of RH pseudo-observations (Z$^{\text{SOP1}}$) than RASTA wind data (V$^{\text{SOP1}}$). Consequently, the ZV$^{\text{SOP1}}$ experiment inherits more from the benefits of the Z$^{\text{SOP1}}$ experiment than from the disadvantages of the V$^{\text{SOP1}}$ experiment in the humidity analysis.

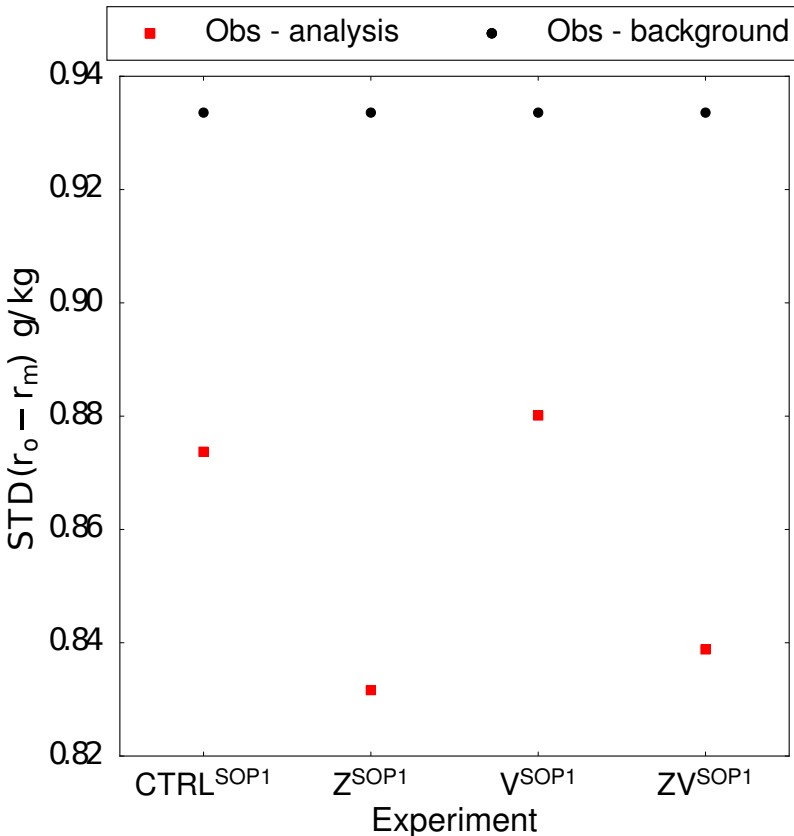

**Figure 11.** Standard deviation (g/kg) *of the water vapour mixing ratio differences between in-flight humidity measurements $r_o$ and the analyses $r_m$* for the different experiments (CTRL[SOP1], Z[SOP1], V[SOP1] and ZV[SOP1]) over the 24 analyses. The standard deviation differences between the measurements and the background state is also shown by the black data points (N=6307).

## 6.3 Rainfall forecast evaluation

Forecast scores are now validated using the rain-gauge network available from the HyMeX database (doi: 10.6096/MISTRALS-HyMeX.904). The rain-gauge measurement locations are indicated by the blue markers in Figure 1. For the comparisons, model outputs are interpolated to the rain-gauge station locations using a linear interpolation. Heidke Skill Score (HSS) is calculated
5   for the 6-h accumulated rainfall forecasts for the CTRL[SOP1] and the three RASTA experiments (Z[SOP1], V[SOP1] and ZV[SOP1]). HSS is calculated for the 32 assimilation cases in which RASTA data have been assimilated. Figure 12 represents the mean HSS differences of the 6-h accumulated rainfall forecasts between the RASTA and the CTRL[SOP1] experiment, as a function of the rainfall accumulation threshold (mm). The bootstrap confidence intervals are also shown for each threshold. They are quite large because HSS has only been calculated over 32 cases. The impact of the assimilation of RASTA wind data is positive if
10  the differences are above 0.

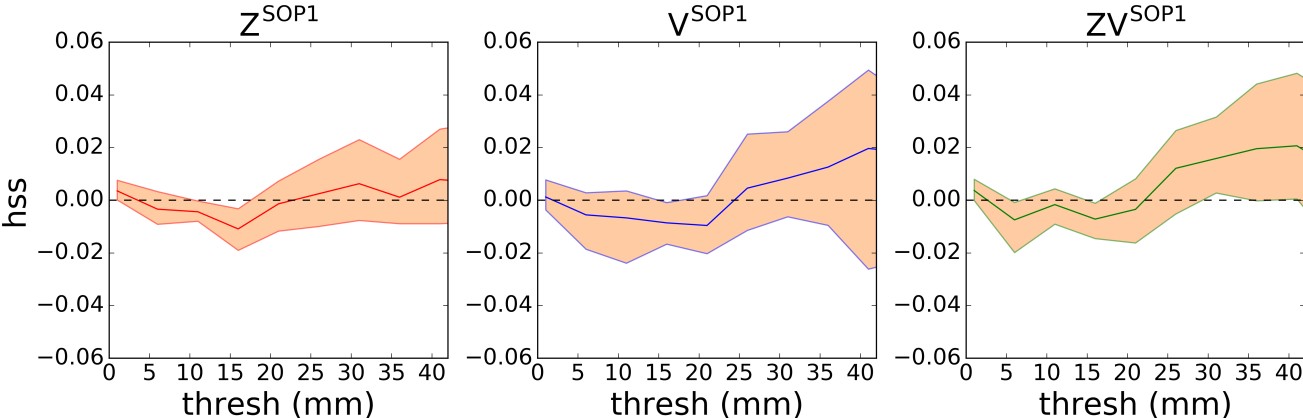

**Figure 12.** Differences in the average HSS of the 6-h cumulated precipitation forecasts versus rain gauge measurements, between the three RASTA experiments and the CTRL$^{SOP1}$ experiment (from left to right: Z$^{SOP1}$, V$^{SOP1}$ and ZV$^{SOP1}$). Calculations were performed over the 32 runs in which RASTA data were assimilated. The error bars represent the 95% bias-corrected and accelerated (BCa) bootstrap confidence intervals (see Efron and Tibshirani, 1993).

Figure 12 indicates that the benefit of the RH pseudo-observations (Z$^{SOP1}$) is neutral to slightly positive above approximately the threshold 25 mm. Besides, the impact of the assimilation of wind vertical profiles (V$^{SOP1}$) is larger than that of RH pseudo-observations (Z$^{SOP1}$), especially for the larger rainfall accumulation thresholds. This is consistent with the fact that the impact of the RH pseudo-observations is less pronounced than the impact of RASTA wind data as the forecast term increases (see

5    section 6). Similar results were also obtained in prior studies (Pu et al., 2009; Zhao and Jin, 2008; Zhang et al., 2012). Finally, the best results are obtained for the ZV$^{SOP1}$ experiment, which suggests that the accumulated rainfall forecasts benefit more from the assimilation of the W-band reflectivity jointly with RASTA wind data. Similar results were also obtained with other categorical scores (FAR and POD), and for the 9- and 12- rainfall accumulation forecasts.

## 7    Discussions and conclusions

10    The primary objective of this article was to assess the impact of the assimilation of W-band radar reflectivity in a kilometre-scale NWP model, specifically to improve analyses and short-term forecasts of heavy precipitation events in the Mediterranean area. The W-band reflectivity measurements collected by the airborne Doppler W-band radar RASTA during the HyMeX-SOP1 were assimilated into the 3h 3DVar assimilation system of the NWP model AROME. To complement this study, the benefit brought by consistent thermodynamic and dynamic cloud conditions has also been investigated by assimilating separately and

15    jointly the horizontal wind measurements retrieved by RASTA. Results of this study will provide guidance for future observing systems by assessing whether it is more relevant to improve the current technologies towards cloud radars measuring horizontal wind profiles, or only reflectivity profiles. The data assimilation experiments were first conducted for one of the most

significant heavy precipitation events of the HyMeX-SOP1 (IOP7a). Then, to cover a larger number of meteorological situations, the different experiments have been run for the 32 cases in which RASTA data were available during the HyMeX SOP1.

The 1D+3DVar assimilation method, operationally employed to assimilate ground-based precipitation radar data in AROME, has been adapted to assimilate the W-band reflectivity. Vertical profiles of relative humidity are first derived via a 1D Bayesian retrieval, and then used as pseudo-observations in the 3DVar assimilation system of AROME. In order to fully take advantage of the W-band reflectivity in cloudy areas, a bias correction scheme was applied. The error standard deviation $\sigma_o$ was estimated by minimising the standard deviation between the retrieved humidity fields and independent in-situ humidity measurements. Results indicate that the best estimate of the error standard deviation is close to 2 dB. The comparison with in-situ humidity measurements highlighted the ability of the 1D Bayesian method to retrieve humidity field which are in better agreement with completely independent humidity measurements.

After validating the first step of the 1D+3DVar assimilation method, the exergy distance was calculated for each experiment to measure the relative impact of the assimilation of RASTA data on the analyses and the subsequent forecasts. This method allows one to assess the impact of the new observation type on the temperature, surface pressure, kinetic and humidity fields, independently. In particular, this impact study demonstrated that RH pseudo-observations have a larger impact on the humidity, temperature and pressure variables on the analyses, compared to the assimilation of RASTA wind data alone. However, after 1 hour forecast, this rank order is reversed, probably because the forecast system has a short memory of the changes made by the RH pseudo-observations on the humidity in the analysis. This result is consistent with the findings of Storto and Tveter (2009), who employed a similar method to assimilate CPR data on-board the satellite CloudSat in the ALADIN NWP model. The impact on the analyses and forecasts is always larger if the W-band reflectivity is assimilated jointly with RASTA wind data, probably because the two observations complement each other and lead to more consistent thermodynamic and dynamic of cloud or frontal conditions in the initial state. In addition, it has been demonstrated that the impact of the assimilation of RH pseudo-observations and/or RASTA wind data is more important over sea than over land, probably because these areas are poorly covered by the conventional network.

To evaluate the benefits brought by these impacts on the analyses, all assimilation experiments have been compared by calculating the standard deviation between the humidity analysis fields and in-situ humidity measurements. The comparisons demonstrated that the experiment in which RH pseudo-observations are assimilated alone improves the most the analyses in terms of humidity, slightly followed by the experiment in which RASTA wind data are also assimilated jointly.

Generally, results of this study indicate that the W-band reflectivity leads to a slight positive improvement of the rainfall precipitation forecasts. Nonetheless, the impact is even more positive if RASTA wind data are assimilated alone. Finally, the best scores are reached if the W-band reflectivity is assimilated jointly with RASTA wind data. Even though for precipitation Doppler radars and for cyclone studies, similar results were also obtained in prior studies (Zhao and Jin, 2008; Pu et al., 2009;

Zhang et al., 2012; Dong and Xue, 2012). Consequently, the results suggest that the assimilation of the two observations jointly leads to a slight improvement of both moisture initial conditions and precipitation forecasts.

In the future, the impact of the assimilation of the W-band reflectivity will also be investigated for other meteorological situations, such as fog. Indeed, since W-band radar are very sensitive to cloud liquid water, their assimilation in km-scale NWP model should improve fog forecasts. In particular, the lower-cost W-band radar BASTA (Delanoë et al., 2016) will be employed during dedicated field campaigns.

The current 1D+3DVar assimilation method requires to define the error standard deviation for the retrieved RH pseudo-observations. One perspective might be to prescribe observation errors that vary in space. In addition, it is possible that the limited impact of RH pseudo-observations as the forecast term increases is due to the fact that hydrometeors are not initialised: the condensation consumes the moisture which has just been injected in the analysis. In a near future, it will be possible to add the hydrometeor specific contents in the control variables with a flow-dependent component in the background-error covariances. Indeed, an EnVar data assimilation system is currently being developed for the AROME model (Montmerle et al., 2018). The direct assimilation of the W-band reflectivity should be favoured by this future implementation.

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

Zhao, Q. and Jin, Y.: High-resolution radar data assimilation for Hurricane Isabel (2003) at landfall, Bulletin of the American Meteorological Society, 89, 1355, 2008.

*Competing interests.* The authors declare that they have no conflict of interest.

*Acknowledgements.* This work is a contribution to the HyMeX program supported by MISTRALS, ANR IODA-MED Grant ANR-11-BS56-10  0005 and ANR MUSIC Grant ANR-14-CE01-0014. This work was supported by the French national programme LEFE/INSU. The authors acknowledge the DGA (Direction Générale de l'Armement), a part of the French Ministry of Defense, for its contribution to Mary Borderies's PhD. The authors thank SAFIRE for operating the French Falcon 20 research aircraft during HyMeX-SOP1. Pierre Brousseau and Thibaut Montmerle are also acknowledged for their technical support and scientific advises.