# Peer review of "Impact of airborne cloud radar reflectivity data assimilation on kilometre-scale NWP analyses and forecasts of heavy precipitation events"

_Natural Hazards and Earth System Sciences, 2018_

## Referee Comment (RC1) · Anonymous Referee #1 · 7 Dec 2018

The results in this study contribute to our understanding of data assimilation method and the manuscript could be considered for publication following revisions.

Please explain the subscript $i$ in Equation (1).

In the bias correction method, the bias is given by $b = <Z^o - y^{po}>$. However, the innovation is defined as $Z^o - H(x)$ in the cost function $J_{po}$. The authors need to show the histogram (or mean error) of the bias corrected innovation.

Please explain the notation of $r_q^o$ and $r_q^m$ in Figure 3.

---

## Referee Comment (RC2) · Anonymous Referee #2 · 4 Jan 2019

This manuscript explores the benefits of assimilating airborne cloud radar observations (reflectivity and wind) to improve forecasts accuracy during heavy precipitation events. The first part of the paper described the radar observations, the AROME-WMed NWP model as well as the assimilation technique used in this study. The study is using a 1D retrieval to best estimate vertical profile of RH pseudo-observation from the radar reflectivity measurements. These profiles, as well as the wind observation, are then assimilated using a traditional 3DVar scheme. Results from the different experiments are presented to establish the impact of the cloud radar observations. This manuscript

was interesting to read (especially the 1D Bayesian retrieval), and dealt with current problems associated with convective-scale data assimilation. I recommend this paper for publication following some revisions.

General comment(s):

1. At the end of section 2.1, a list of the different weather patterns present during HyMex-SOP1 campaign is presented. I am surprised to see that the authors did not use this information during the impact study (section 6). It would have been very interesting to see how the impact of assimilating the radar observation is affected by the meteorological configuration.

Specific comment(s):

1. P 6 – section 2.2: The authors should tell us a bit more about the assimilation system. For example:

- What is the resolution of the analysis grid?

- Does the 3Dvar scheme is using an IAU?

2. P 6 - section 2.3: It is not clear if the data are thinned vertically?

---

## Author Comment (AC1) · 19 Feb 2019

We thank Reviewer #1 for his/her constructive comments. Our responses are given in red in the attached file, along with the modified version of the article.

Please also note the supplement to this comment:
https://www.nat-hazards-earth-syst-sci-discuss.net/nhess-2018-314/nhess-2018-314-AC1-supplement.zip

---

## Author Comment (AC2) · 19 Feb 2019

We are grateful to Reviewer #2 for his/her constructive comments. Our responses are given in red in the attached file, along with the modified version of the article.

Please also note the supplement to this comment: https://www.nat-hazards-earth-syst-sci-discuss.net/nhess-2018-314/nhess-2018-314-AC2-supplement.zip

---

## Author Response (AR1)

We thank Reviewer #1 for his/her constructive comments.
Our responses are given below in red.

Please explain the subscript I in Equation (1).
We added "subscript $i$ denotes the index of the model profile in the vicinity of the observed profile of reflectivity"

In the bias correction method, the bias is given by b= $Z^o$ – $Z^{PO}$, However the innovation is defined as $Z^o$ – H(x) in the cost function $J_{po}$. The authors need to show the histogram (or mean error) of the bias corrected innovation.

The Bayesian retrieval is not variational, and therefore $Z^o$ – H(x) cannot be defined as an innovation. The fact that we introduced $J_{PO}$ as a cost function was misleading in the first version of the paper. $J_{PO}$ is a weight associated to each column $i$ in the vicinity of the radar. It is a function of the difference between observed and simulated RASTA reflectivities. The text " in equation 2" has been modified by "in equation 2".

The bias correction was not calculated using $Z^o$ – H(x). Indeed, grid-to-grid comparisons require a perfect spatial and temporal match between observations and forecasts, which is rarely the case for high-resolution NWP models, and especially when convective systems are considered. By construction, the Bayesian retrieval allows to shift a pattern that was well simulated by the model, but at a wrong location. In order to remove positional errors, which are not gaussian, from the bias correction, it has been decided to define the bias by b= $Z^o$ – $Z^{PO}$, instead of $Z^o$ – H(x).

The histogram and mean error of the bias-corrected reflectivity pseudo-observations are now shown in Figure 3. We added some explanations in the text in section 3.2:
*"The effect of the bias correction is shown in Figure 3, in which Contoured Frequency by Altitude Diagram (CFAD) of the differences between the observed reflectivity and the bias-corrected reflectivity pseudo-observations are shown for a $\sigma_o$ of 2 dB. The new bias is indicated by the black line. Figure 3 demonstrates that, after applying the bias correction in Equation 2, the residual bias is close to 0 dB except above an altitude of approximately 10 km, which is probably due to the smaller number of points used to calculate the bias correction. As explained by Janisková (2015), the use of additional predictors, such as temperature or hydrometeor contents, could lead to an improvement in the bias correction at higher altitude."*

Please explain the notation of ro_q and r_m_q in Figure 3
We changed the two notations for $r_o$ and $r_m$ in Figures 3 and 10 (now Figures 4 and 11). $r_o$ denotes in-flight water vapour mixing ratio measurements. $r_m$ is always the water vapour mixing ratio from the model.

In Figure 4, $r_m$ denotes the water vapor mixing ratio retrieved using the 1D Bayesian method. The red curve indicates the standard deviation errors (and biases on the left panel) between $r_o$ and $r_m$. Similarly, the black curve indicates the standard deviation errors (and biases) between $r_o$ and water vapour mixing ratio from the background.

In Figure 11, $r_m$ indicates the water vapour mixing ratio from the analyses.

We thank Reviewer #2 for his/her constructive comments.
Our responses are given below in red.

General comment(s):
1. At the end of section 2.1, a list of the different weather patterns present during HyMex-SOP1 campaign is presented. I am surprised to see that the authors did not use this information during the impact study (section 6). It would have been very interesting to see how the impact of assimilating the radar observation is affected by the meteorological configuration.

The index used at the end of section 2.1 is not employed to characterise the different weather patterns, but only the vertical columns observed by RASTA during HyMeX-SOP1. Indeed, all our case studies are convective ones. However, some (72.6 %) of the data collected by RASTA were collected in the stratiform parts of the convective cells. To make this clearer, we modified *"… in clear sky (13.1%)* **conditions"** by *… in clear sky (13.1%)* **columns"**.

Therefore, because we only have convective case studies, we could not study how the impact of the assimilation of RASTA data is affected by the meteorological configuration. However, we conducted a similar work to study if the impact is more pronounced over land than over sea (see section 6.1.3), which highlighted that the impact is generally larger over sea than over land.

Specific comment(s):
1. P 6 – section 2.2: The authors should tell us a bit more about the assimilation system. For example:
- What is the resolution of the analysis grid?

The resolution of the analysis grid is the same as the one used in AROME-WMed: 2.5 km.
We added this information in the text in section 2.2 *"The resolution of the analysis grid is the same as that of AROME-WMed".*

- Does the 3Dvar scheme is using an IAU?

The IAU was not used in the 3DVar data assimilation scheme of our version of Arome that has a 3-h update cycle. The use of the IAU has been evaluated with this version by Brousseau (2012). He has demonstrated that the IAU reduces the spin-up, but does not improve the forecast performance and can even lead to forecast degradations.

In the latest versions of Arome that use 1-h update cycles, the IAU is now used once or twice a day, but merely because of production constraints and not for its filtering properties. In this particular configuration, the IAU does not alter the performance of the system (Brousseau et al. 2016, §4.2).

We added a sentence in section 2.2:
*"Following the results of Brousseau (2012), the Incremental Analysis Update (IAU, Bloom et al., 1996) is not used for the 3DVar assimilation scheme."*

Brousseau, 2012: Propagation of observed information into the AROME data assimilation and atmospheric model, PhD thesis, Université de Toulouse III – Paul Sabatier

Brousseau, P. , Seity, Y. , Ricard, D. and Léger, J. (2016), Improvement of the forecast of convective activity from the AROME-France system. Q.J.R. Meteorol. Soc., 142: 2231-2243. doi:10.1002/qj.2822

2. P 6 - section 2.3: It is not clear if the data are thinned vertically?

The relative humidity pseudo-observations are not thinned vertically.

Indeed, Jacques and Zawadzki (2014) showed that "Data thinning may only alleviate the errors caused by correlation misrepresentations *on the condition that background errors are sufficiently correlated.*". Brousseau et al. (2011) (their Figures 5 and 6) demonstrated that vertical background error covariances are less marked than the horizontal ones. Therefore, in the AROME model the observations are not thinned vertically.

We added in section 2.3 *"The data are not thinned vertically because the vertical forecast error covariances are less marked than the horizontal ones (Brousseau et al. 2011) and it is thus not useful to apply any thinning in that case (Jacques and Zawadzki 2014)."*

Brousseau P, Berre L, Bouttier F, Desroziers G. 2011. Background-error covariances for a convective-scale data-assimilation system : AROME- France 3DVar. Quart. J. Roy. Meteor. Soc. 137: 409–422.

Jacques, D., I. Zawadzki, 2014: The Impacts of Representing the Correlation of Errors in Radar Data Assimilation. Part I: Experiments with Simulated Background and Observation Estimates. Monthly Weather Review, 142(11), 3998-4016, DOI: 10.1175/mwr-d-14-00104.1.